# Wrapped Gaussian on the manifold of Symmetric Positive Definite Matrices

**Thibault de Surrel** [1]   **Fabien Lotte** [2]   **Sylvain Chevallier** [3]   **Florian Yger** [4]

## Abstract

Circular and non-flat data distributions are prevalent across diverse domains of data science, yet their specific geometric structures often remain underutilized in machine learning frameworks. A principled approach to accounting for the underlying geometry of such data is pivotal, particularly when extending statistical models, like the pervasive Gaussian distribution. In this work, we tackle those issue by focusing on the manifold of symmetric positive definite (SPD) matrices, a key focus in information geometry. We introduce a non-isotropic wrapped Gaussian by leveraging the exponential map, we derive theoretical properties of this distribution and propose a maximum likelihood framework for parameter estimation. Furthermore, we reinterpret established classifiers on SPD through a probabilistic lens and introduce new classifiers based on the wrapped Gaussian model. Experiments on synthetic and real-world datasets demonstrate the robustness and flexibility of this geometry-aware distribution, underscoring its potential to advance manifold-based data analysis. This work lays the groundwork for extending classical machine learning and statistical methods to more complex and structured data.

## 1. Introduction

When dealing with complex data, modeling them as lying on a manifold often provides a powerful solution (Sanborn et al., 2024; Jo & Hwang, 2024). However, classical Euclidean probability distributions fail to capture the intrinsic geometry of the underlying manifold. This limitation necessitates adapting the choice of probability distributions to respect the manifold's structure. In this work, we propose a

solution by leveraging the exponential map to wrap probability distributions defined in the Euclidean tangent space onto the manifold itself. This approach yields a *wrapped distribution*, intrinsically aligned with the manifold's geometry. Wrapped distributions have found applications across many domains, such as embedding single-cell RNA data (Ding & Regev, 2021), analyzing wave patterns (Jona-Lasinio et al., 2012), recognizing video and image features (Turaga et al., 2011), modeling Gaussian processes on manifolds (Mallasto & Feragen, 2018; Liu et al., 2024) or doing statistics on measurements of orientations (Lopez-Custodio, 2024).

In this paper, we focus on wrapping Gaussian distributions on the Riemannian manifold of *Symmetric Positive Definite* (SPD) matrices equipped with the *Affine Invariant Riemannian Metric* (Pennec, 2020). Gaussian distributions are a cornerstone of machine learning and statistics (see p.102 of Casella & Berger 2001) due to their ubiquity and theoretical underpinnings, such as the Central Limit Theorem (CLT) (Section 5.4 of Wasserman 2004), which ensures that Gaussian distributions naturally arise in many scenarios. We will see that we are able to extend the CLT to wrapped Gaussians, providing a theoretical justification for their study. SPD matrices, which form the Riemannian manifold $\mathcal{P}_d$, are pivotal in numerous applications, including *Diffusion Tensor Imaging* (Pennec, 2020), *Brain-Computer Interfaces* (BCI) (Lotte et al., 2018), *process control* (Willjuice Iruthayarajan & Baskar, 2010), and *video processing* (Tuzel et al., 2008). The inherent Riemannian geometry of $\mathcal{P}_d$ necessitates adopting methods that respect its manifold structure when analyzing SPD data.

This paper is organized as follows: in Section 3, we introduce the Riemannian geometry of $\mathcal{P}_d$. Next, we define wrapped Gaussians on $\mathcal{P}_d$ and explore their theoretical properties in Section 4. In Section 5, we develop a Maximum Likelihood Estimator for parameter estimation and validate it with synthetic experiments. Building on this foundation, Section 6 revisits existing classifiers on $\mathcal{P}_d$ through a probabilistic lens and introduces novel classifiers based on wrapped Gaussians. Finally, we evaluate these classifiers with synthetic and real-world datasets in Section 6.4.

---

[1]LAMSADE, CNRS, PSL Univ. Paris-Dauphine, France [2]Inria center at the University of Bordeaux / LaBRI, France [3]TAU, LISN, University Paris-Saclay, France [4]LITIS, INSA Rouen-Normandy, France. Correspondence to: Thibault de Surrel <thibault.de-surrel@lamsade.dauphine.fr>.

*Proceedings of the 42nd International Conference on Machine Learning*, Vancouver, Canada. PMLR 267, 2025. Copyright 2025 by the author(s).

## 2. Related works

Other works have already tried to extend the Gaussian distribution to a Riemannian manifold. Said et al. (2018) propose an isotropic Gaussian on a Riemannian Symmetric Space $\mathcal{M}$ defined using a center of mass $\bar{y} \in \mathcal{M}$ and a scaling factor $\sigma > 0$. In our work, we are looking for a more complex model, requiring a non-isotropic distribution with some preferred directions. A non-isotropic Gaussian distribution on a manifold has been proposed in Pennec (2006), in which the authors use the characterization of the Gaussian distribution as the distribution that maximizes entropy given a mean and a covariance matrix (theorem 13.2.2 of Kagan et al. 1973). However, sampling this distribution leads to computational difficulties, the normalizing constant cannot be computed explicitly and in the case of a full covariance matrix, the estimator of the parameters becomes problematic.

Wrapped distributions have first been studied in directional statistics (Mardia & Jupp, 2000), on a circle (Collett & Lewis, 1981) or on a sphere (Hauberg, 2018). Wrapped Gaussians have also been instantiated on hyperbolic spaces, first by Nagano et al. (2019) and then by Mathieu et al. (2019) and Cho et al. (2022). They mainly use it as the distribution of the latent space of a Variational Autoencoder which is trained to learn the distribution. Apart from the manifold, another difference with our approach is that they wrap the Gaussian using a composition of the exponential map with parallel transport where we will only use the exponential map. Wrapped distributions have also been studied on more general classes of Riemannian manifolds. For example, Galaz-Garcia et al. (2022) define wrapped distributions on homogeneous Riemannian manifolds. A major difference with our work is that they use a volume preserving map to push-forward the density from the tangent space to the manifold, leading to a simpler expression of the density, without any volume change term. In Chevallier et al. (2022) and Chevallier & Guigui (2020), the authors work on general symmetric spaces and mainly study the Jacobian determinant of the exponential map, first in a general setting and then on different examples (Grassmannians, pseudohyperboloids and special Euclidean group). Unlike our work, they consider the distribution on the tangent space to always be centered, where we consider a more general setting by allowing $\mu \neq 0$. To estimate the parameters of their wrapped Gaussians, they use moment estimation. Finally, in Troshin & Niculae (2023), they define a more general wrapped Gaussian, the $\beta-$Gaussian that has a compact support.

In the following, we propose a wrapped Gaussian on the manifold of SPD matrices that is not centered on the tangent space. After deriving some theoretical properties, we show that our wrapped Gaussian can be used in practice, showcasing the estimation of the parameters from a finite number of samples. Finally, we use our wrapped Gaussian to build a framework that unifies and generalizes classification on SPD matrices, and propose new classifiers. This application shows the potential of our wrapped Gaussian to become a generic, flexible and powerful tool for manifold-based data analysis.

## 3. How to deal with SPD matrices ?

### 3.1. The Riemannian geometry of SPD matrices

The set of $d \times d$ *symmetric, positive definite* (SPD) matrices, denoted $\mathcal{P}_d$ is defined as follows:

$$\mathcal{P}_d = \{p \in \mathbb{R}^{d \times d} \mid p^\top = p \text{ and } \forall x \in \mathbb{R}^d \setminus \{0\}, x^\top p x > 0\}.$$

This set is convex and open in the set of $d \times d$ symmetric matrices $\mathcal{S}_d$ and thus, it is a *manifold* of dimension $d(d+1)/2$. For all $p \in \mathcal{P}_d$, the tangent space $T_p\mathcal{P}_d$ at $p$ can be identified with $\mathcal{S}_d$. Moreover, for $p \in \mathcal{P}_d$ one can define an inner product on the tangent space $T_p\mathcal{P}_d$ at $p$ by:

$$\forall u, v \in T_p\mathcal{P}_d, \ \langle u, v \rangle_p = \mathrm{tr}(p^{-1}up^{-1}v). \tag{1}$$

This inner product is called the *Affine Invariant Riemannian Metric* (AIRM) (Pennec, 2020) as, if $a$ is an invertible matrix, one has $\langle aua^\top, ava^\top \rangle_{apa^\top} = \langle u, v \rangle_p$. Once endowed with this metric, $\mathcal{P}_d$ is a complete connected Riemannian manifold of non-positive curvature (see Appendix I of Criscitiello & Boumal 2023). It is therefore a *Hadamard manifold* (Shiga, 1984). Using the Cartan–Hadamard theorem (theorem 12.8 of Lee 2018) one has that $\mathcal{P}_d$ is diffeomorphic to $\mathbb{R}^{d(d+1)/2}$ through the exponential map $\mathrm{Exp}_p \colon T_p\mathcal{P}_d \simeq \mathbb{R}^{d(d+1)/2} \to \mathcal{P}_d$. Another consequence of the completeness of $\mathcal{P}_d$ is that each pair of points $(p, q) \in \mathcal{P}_d^2$ can be connected by a unique minimizing geodesic whose length defines a distance on $\mathcal{P}_d$. This AIRM distance, between $p$ and $q$ is given by:

$$\delta(p, q) = \| \log(p^{-1/2}qp^{-1/2}) \|_F \tag{2}$$

where $\log$ is the matrix logarithm and $\| \cdot \|_F$ the Frobenius norm. Other useful tools of Riemannian geometry that will be used in the following are the exponential map $\mathrm{Exp}_p \colon T_p\mathcal{P}_d \to \mathcal{P}_d$ and its inverse, the logarithm map $\mathrm{Log}_p \colon \mathcal{P}_d \to T_p\mathcal{P}_d$. For $p, q \in \mathcal{P}_d$ and $u \in T_p\mathcal{P}_d$, those maps are given by the following expressions (see chapter 6 of (Bhatia, 2007)):

$$\begin{aligned} \mathrm{Exp}_p(u) &= p^{1/2} \exp(p^{-1/2}up^{-1/2})p^{1/2}, \\ \mathrm{Log}_p(q) &= p^{1/2} \log(p^{-1/2}qp^{-1/2})p^{1/2}. \end{aligned} \tag{3}$$

Finally, when $\mathcal{P}_d$ is equipped with the AIRM metric given at Equation (1), the Riemannian volume element at $p = [[p_{ij}]] \in \mathcal{P}_d$ is given by (see Section 4.1.3 of Terras 1988):

$$\mathrm{dvol}(p) = \det(p)^{-(d+1)/2} \prod_{i \leq j} \mathrm{d}p_{ij} \tag{4}$$

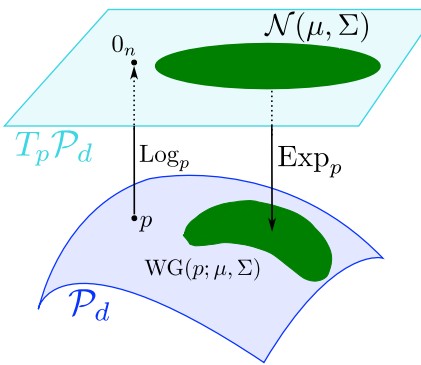

*Figure 1.* An illustration of a wrapped Gaussian $\mathrm{WG}(p; \mu, \Sigma)$.

---

**Algorithm 1** Sampling from a Wrapped Gaussian $\mathrm{WG}(p; \mu, \Sigma)$

---

**Require:** $p \in \mathcal{P}_d, \mu \in \mathbb{R}^{d(d+1)/2}, \Sigma \in \mathcal{P}_{d(d+1)/2}$
 1: Sample $\mathbf{t} \sim \mathcal{N}(\mu, \Sigma)$
 2: Compute $X \leftarrow \mathrm{Exp}_p(\mathrm{Vect}_p^{-1}(\mathbf{t}))$
 3: Return $X \sim \mathrm{WG}(p; \mu, \Sigma)$

---

a given wrapped Gaussian can be parametrized by different sets of parameters, leading to an equivalence relation. In the following, we will denote $\Theta = \mathcal{P}_d \times \mathbb{R}^{d(d+1)/2} \times \mathcal{P}_{d(d+1)/2}$ the space of parameters of the wrapped Gaussian which is a product manifold.

### 4.1. The definition

To define a *wrapped Gaussian*, we start with a classical Euclidean Gaussian random variable in $\mathbb{R}^{d(d+1)/2}$ and push it on the manifold $\mathcal{P}_d$ using the exponential map as follows:

**Definition 4.1** (Wrapped Gaussian). Let $p \in \mathcal{P}_d$, $\mu \in \mathbb{R}^{d(d+1)/2}$ and $\Sigma \in \mathcal{P}_{d(d+1)/2}$. A random variable $X$ on $\mathcal{P}_d$ follows a wrapped Gaussian denoted $\mathrm{WG}(p; \mu, \Sigma)$ if

$$X = \mathrm{Exp}_p(\mathrm{Vect}_p^{-1}(\mathbf{t})), \ \mathbf{t} \sim \mathcal{N}(\mu, \Sigma).$$

A wrapped Gaussian is illustrated in Figure 1. This gives us a very simple algorithm to sample a wrapped Gaussian $\mathrm{WG}(p; \mu, \Sigma)$, since it is simply based on the sampling of $\mathcal{N}(\mu, \Sigma)$ in $T_p\mathcal{P}_d$ and the computation of $\mathrm{Vect}_p^{-1}$ and $\mathrm{Exp}_p$ that has closed-form formulas. We provide a sampling algorithm at Algorithm 1. Moreover, we can rewrite this definition using a *push-forward* (see definition 2.1 of Peyré & Cuturi (2020), Section 3.6 of Bogachev (2007) or Appendix B):

$$\mathrm{WG}(p; \mu, \Sigma) = (\mathrm{Exp}_p \circ \mathrm{Vect}_p^{-1}) \# \mathcal{N}(\mu, \Sigma) \quad (5)$$

where $\#$ denotes the push-forward operator.

Let us comment on the different parameters: $p$ gives us a tangent plan from which the Gaussian is wrapped, so it locates the Gaussian on the manifold. $\mu$ is the mean of the Gaussian in the tangent space $T_p\mathcal{P}_d$. As $\mathcal{P}_d$ and $T_p\mathcal{P}_d$ are in bijection through the exponential map, $p$ and $\mu$ play a symmetric role that will be unveiled in Section 4.4. Finally, $\Sigma$ is the covariance matrix between the entries of the SPD matrices. In the special case where the SPD matrices are covariance matrices, $\Sigma$ models the covariance of the covariances, therefore, it can be seen as a fourth order moment.

### 4.2. The density of a wrapped Gaussian

Now that we have defined the wrapped Gaussian, we give its density using the push-forward definition of $\mathrm{WG}(p; \mu, \Sigma)$ given in Equation (5) (see Theorem B.2 in Appendix B)

---

where $\mathrm{d}p_{ij}$ is the Lebesgue measure on $\mathbb{R}$. The volume $\mathrm{dvol}$ is also invariant under congruence by invertible matrices.

*Remark* 3.1. Other metrics could be used on $\mathcal{P}_d$, such as the *Log-Euclidean metric* (Arsigny et al., 2005) or the *S-divergence* metric (Sra, 2015). However, the AIRM metric is widely used in the literature and has many properties that make it suitable for our work (such as the affine invariance). For more details on the different metrics on $\mathcal{P}_d$, we refer the reader to Chevallier et al. (2021) and Thanwerdas (2022).

### 3.2. The vectorization of the tangent spaces

As described in Section 3.1, the tangent space $T_p\mathcal{P}_d$ at point $p \in \mathcal{P}_d$ is identified with the space of $d \times d$ symmetric matrices $\mathcal{S}_d$ which is isomorphic to $\mathbb{R}^{d(d+1)/2}$. We define such an isomorphism called the *vectorization* as follows:

**Definition 3.2** (Vectorization). We start by defining the vectorization at identity for a symmetric matrix $u = [[u_{ij}]]$:

$$\mathrm{Vect}_{I_d} \colon u \in T_{I_d}\mathcal{P}_d \mapsto (u_{11}, \sqrt{2}u_{12}, u_{22}, \sqrt{2}u_{13}, \sqrt{2}u_{23}, u_{33},$$
$$\ldots, \sqrt{2}u_{d-1,d}, u_{dd}) \in \mathbb{R}^{d(d+1)/2}$$

Then, for $p \in \mathcal{P}_d$, we define the vectorization at $p$:

$$\mathrm{Vect}_p \colon u \in T_p\mathcal{P}_d \mapsto \mathrm{Vect}_{I_d}(p^{-1/2}up^{-1/2}).$$

One of the important property of $\mathrm{Vect}_p$ is that it is an isometry between $(T_p\mathcal{P}_d, \langle \cdot, \cdot \rangle_p)$ and $(\mathbb{R}^{d(d+1)/2}, \langle \cdot, \cdot \rangle_2)$. More information on this vectorization can be found in Section 3.3.3.3. of Pennec (2020) or in Appendix A.

## 4. Wrapped Gaussian on the manifold of SPD matrices

In this work, our objective is to define a non-isotropic Gaussian on the manifold of SDP matrices $\mathcal{P}_d$. In this section, we define a *wrapped Gaussian* through the way it will be sampled. We will also give the density of a wrapped Gaussian and, we will show that, unlike usual probability distributions,

---

**Theorem 4.2.** *The density $f_{p;\mu,\Sigma}$ of the wrapped Gaussian $\mathrm{WG}(p;\mu,\Sigma)$ exists and is:*

$$\forall x \in \mathcal{P}_d, \ f_{p;\mu,\Sigma}(x) = \frac{g_{\mu,\Sigma}(\mathrm{Vect}_p(\mathrm{Log}_p(x)))}{|J_p(\mathrm{Log}_p(x))|} \quad (6)$$

*where $g_{\mu,\Sigma}$ is the density of the multivariate Gaussian $\mathcal{N}(\mu,\Sigma)$ and $J_p(\cdot) = \det(\mathrm{d}\,\mathrm{Exp}_p(\cdot))$ is the Jacobian determinant of the exponential map $\mathrm{Exp}_p$.*

The Jacobian determinant of the exponential map describes the volume change induced by the identification of $T_p\mathcal{P}_d$ and $\mathcal{P}_d$ by the exponential map $\mathrm{Exp}_p$. To compute the density explicitly, one needs to compute the Jacobian $J_p(u)$, which has a closed form expression for the manifold $\mathcal{P}_d$ (see Section 5.3 of Chevallier & Guigui (2020) or Appendix C).

**Proposition 4.3.** *The Jacobian determinant of the exponential map at the identity $\mathrm{Exp}_{I_d}$ is:*

$$\forall u \in T_{I_d}\mathcal{P}_d, \ J_{I_d}(u) = 2^{d(d-1)/2} \prod_{i<j} \frac{\sinh\left(\frac{\lambda_i(u)-\lambda_j(u)}{2}\right)}{\lambda_i(u)-\lambda_j(u)}$$

*where the $\lambda_i(u)$ are the eigenvalues of $u$. Then, one can use the previous formula to compute the Jacobian determinant of the exponential map at any point $p \in \mathcal{P}_d$:*

$$\forall u \in T_p\mathcal{P}_d, \ J_p(u) = J_{I_d}(p^{-1/2}up^{-1/2}).$$

It should be noted that, unlike the wrapped Gaussians defined in Galaz-Garcia et al. (2022), Troshin & Niculae (2023) and Nagano et al. (2019), we do not restrict ourselves to a centered multivariate Gaussian $\mathcal{N}(0,\Sigma)$ on the tangent space $T_p\mathcal{P}_d$. In our case, we allow the wrapped Gaussian to have an extra parameter $\mu$, thus extending the flexibility and applicability of the model. Having a noncentered distribution on the tangent space $T_p\mathcal{P}_d$ leads to new considerations that will be discussed in Section 4.4.

*Remark* 4.4. In this work, we focus on extending the multivariate Gaussian to a Riemannian setting. With no extra difficulty, one could wrap an *Elliptically Contoured distribution* (EC) on $\mathcal{P}_d$ (chapter 6 of Johnson (1987) or Delmas et al. 2024). We give more detail on this in Appendix F.

### 4.3. Some properties of wrapped Gaussians

Let us now give some properties of the wrapped Gaussians. We start by a rescaling property.

**Proposition 4.5.** *Let $X \sim \mathrm{WG}(I_d; 0_{d(d+1)/2}, I_{d(d+1)/2})$ and let $(p,\mu,\Sigma) \in \Theta$. There exists a transformation of $X$ denoted $\Psi$ such that*

$$\Psi(X) \sim \mathrm{WG}(p;\mu,\Sigma).$$

*Thus, the wrapped Gaussian $\mathrm{WG}(I_d; 0_{d(d+1)/2}, I_{d(d+1)/2})$ is a building block of the wrapped Gaussians.*

One can find a proof of this result, as well as an explicit expression for $\Psi$ in Appendix D.

Then, we give a wrapped version of the multivariate *Central Limit Theorem* (CLT) (Theorem 5.12 of Wasserman 2004) for the manifold $\mathcal{P}_d$. For this, we define the logarithmic product introduced in Arsigny et al. (2006):

**Definition 4.6** (Logarithmic product). Let $p, q \in \mathcal{P}_d$. The logarithmic product of $p$ and $q$ is defined as:

$$p \odot q = \exp(\log p + \log q).$$

Equipped with this logarithmic product, $(\mathcal{P}_d, \odot)$ forms a commutative group that is isomorphic to $(\mathcal{S}_d, +)$[1]. One can generalize this notation to the sum of $n$ SPD matrices $p_1,\ldots,p_n$ as $\bigodot_{i=1}^{n} p_i = p_1 \odot \cdots \odot p_n$. Using this logarithmic product, we can state the following theorem:

**Theorem 4.7** (Wrapped CLT). *Let $(X_i)_{i\in\mathbb{N}^*}$ be a sequence of i.i.d. random variables on $\mathcal{P}_d$. We suppose that the sequence $(\mathrm{Vect}_{I_d}(\mathrm{Log}_{I_d}(X_i)))_{i\in\mathbb{N}^*}$ of random variables on $\mathbb{R}^{d(d+1)/2}$ admits a finite second order moment. We denote by $\mu$ the mean and by $\Sigma$ the covariance matrix of $\mathrm{Vect}_{I_d}(\mathrm{Log}_{I_d}(X_1))$. Then,*

$$\left(\bigodot_{i=1}^{n}(X_i \odot m^{-1})\right)^{\frac{1}{\sqrt{n}}} \xrightarrow[n\to\infty]{d} \mathrm{WG}(I_d; 0, \Sigma)$$

*where $\xrightarrow[n\to\infty]{d}$ denotes the convergence in distribution and where $m = \mathrm{Exp}_{I_d}(\mathrm{Vect}_{I_d}^{-1}(\mu))$.*

This theorem shows the interest of wrapped Gaussians, as they naturally appear in the limit of a logarithmic product of random SPD matrices. The proof of this theorem can be found in Appendix E. One can note that by generalizing the logarithmic product defined in Definition 4.6 to another tangent space $T_p\mathcal{P}_d$, one can extend the limit to $\mathrm{WG}(p; 0, \Sigma)$.

We can also give information on the mean of $\mathrm{WG}(p;\mu,\Sigma)$ in the special case of $\mu = 0$ i.e. when the distribution on the tangent space $T_p\mathcal{P}_d$ is centered. We recall from definition 3 of Pennec (2019) that a mean, or exponential barycenter, of a probability distribution $\alpha$ on $\mathcal{P}_d$ is defined as a point $\bar{p} \in \mathcal{P}_d$ satisfying $\int_{\mathcal{P}_d} \mathrm{Log}_{\bar{p}}(x)\mathrm{d}\alpha(x) = 0$. For wrapped Gaussians, one has the following result:

**Proposition 4.8.** *A mean of $\mathrm{WG}(p; 0, \Sigma)$ is $p$.*

The proof is straightforward from the definition of the wrapped Gaussian $\mathrm{WG}(p; 0, \Sigma)$.

---

[1]More information on the properties of this logarithmic product can be found in Arsigny et al. (2006).

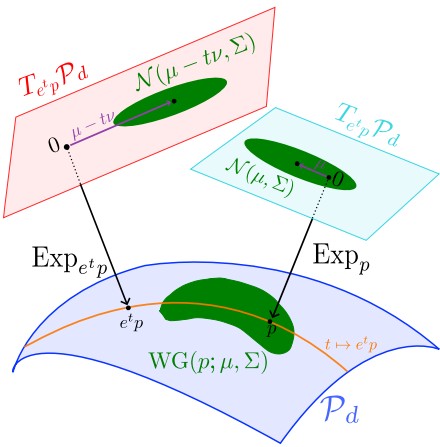

*Figure 2.* Illustration of the equivalence between two wrapped Gaussians given by Proposition 4.9.

## 4.4. An equivalence relation between wrapped Gaussians

A key property of the Euclidean Gaussian $\mathcal{N}(\mu, \Sigma)$ is its *identifiability*, meaning that the map $(\mu, \Sigma) \mapsto \mathcal{N}(\mu, \Sigma)$ is one-to-one (definition 1.5.2 of Lehmann & Casella 1998). The notion of identifiability is important when one wants to learn the parameters of a distribution from a finite set of samples as we will do in Section 5. If we consider a model based on the wrapped Gaussian, we lose this property as we have the following proposition, illustrated in Figure 2:

**Proposition 4.9.** *Let $(p, \mu, \Sigma) \in \Theta$ and $t \in \mathbb{R}$. One has that $\mathrm{WG}(p; \mu, \Sigma)$ and $\mathrm{WG}(e^t p; \mu - t\nu, \Sigma)$ are equal where*
$$\nu = \mathrm{Vect}_p(p) = (\underbrace{1, \cdots, 1}_{d \text{ ones}}, \underbrace{0, \cdots, 0}_{d(d-1)/2 \text{ zeros}}) \in \mathbb{R}^{d(d+1)/2}.$$

All the proofs of this section can be found in Appendix G. One can verify using Equation (3), that $t \mapsto e^t p$ is the geodesic $\gamma \colon t \mapsto \mathrm{Exp}_p(tp)$ in $\mathcal{P}_d$ starting at point $p$ and with initial velocity $p$ (as $p$ is a symmetric matrix, it also belongs to $T_p \mathcal{P}_d \simeq \mathcal{S}_d$). Moreover, the map $t \mapsto \mu - t\nu = \mu - t\,\mathrm{Vect}_p(p)$ is also the geodesic in $\mathbb{R}^{d(d+1)/2}$ with initial point $\mu$ and initial velocity $-\nu = -\,\mathrm{Vect}_p(p)$. Therefore, when $p$ is pushed in one "direction" (initial velocity $p$), $\mu$ is pushed in the opposite "direction" (initial velocity $-\,\mathrm{Vect}_p(p)$).

A wrapped Gaussian can thus be represented by several sets of parameters, so we define an *equivalence relation* between sets of parameters that define the same wrapped Gaussian:

**Definition 4.10.** Let $\theta_\alpha = (p_\alpha, \mu_\alpha, \Sigma_\alpha) \in \Theta$ and $\theta_\beta = (p_\beta, \mu_\beta, \Sigma_\beta) \in \Theta$ be two sets of parameters. Then, $\theta_\alpha$ and $\theta_\beta$ are equivalent, which we denote by $\theta_\alpha \cong \theta_\beta$, if they define the same wrapped Gaussian i.e.

$$\mathrm{WG}(p_\alpha; \mu_\alpha, \Sigma_\alpha) = \mathrm{WG}(p_\beta; \mu_\beta, \Sigma_\beta).$$

We denote by $[\theta_\alpha]$ the equivalence class of $\theta_\alpha$:

$$[\theta_\alpha] = \{\theta = (p', \mu', \Sigma') \mid \theta \cong \theta_\alpha\}.$$

Using Proposition 4.9, one has the immediate corollary:

**Corollary 4.11.** *Let $\theta_\alpha = (p_\alpha, \mu_\alpha, \Sigma_\alpha) \in \Theta$ and $\theta_\beta = (p_\beta, \mu_\beta, \Sigma_\beta) \in \Theta$. If there exists $t \in \mathbb{R}$ such that $p_\beta = e^t p_\alpha, \mu_\beta = \mu_\alpha + t\,\mathrm{Vect}_{p_\alpha}(p_\alpha)$ and $\Sigma_\beta = \Sigma_\alpha$, then $\theta_\alpha \cong \theta_\beta$.*

*Remark* 4.12. All equivalence classes do not contain a wrapped Gaussian that is centered on the tangent space ($\mu = 0$). Let us consider a wrapped Gaussian $\mathrm{WG}(p; \mu, \Sigma)$. Then, the equivalent wrapped Gaussians are of the form $\mathrm{WG}(e^t p; \mu - t\nu, \Sigma)$ for $t \in \mathbb{R}$. If $\mu$ and $\nu$ are aligned i.e., there exists $\tilde{t} \in \mathbb{R}$ such that $\mu = \tilde{t}\nu$, then the equivalence class contains a wrapped Gaussian with $\mu = 0$. However, when $\mu$ and $\nu = (1, \cdots, 1, 0, \cdots, 0)$ are not aligned (for example, take $\mu = \nu + (1, \cdots, 0) = (2, 1, \cdots, 1, 0, \cdots, 0)$), then there exists no $t \in \mathbb{R}$ such that $\mu = t\nu$ and the equivalence class does not contain a wrapped Gaussian with $\mu = 0$. Therefore, allowing $\mu \neq 0$ increases the expressiveness of the model.

Once we have defined an equivalence class $[\theta]$ of parameters that define the same wrapped Gaussian, it is natural to define a representative of $[\theta]$. We define it as follows:

**Definition 4.13** (Representative of an equivalence class)**.** We choose as representative of the class $[\theta]$, the tuple of parameters $\theta^{\min} = (p^{\min}, \mu^{\min}, \Sigma^{\min})$ such that $\mu^{\min}$ is minimal in the sens of $\| \cdot \|_2$. We call it the *minimal representative*.

One is able, given a tuple of parameters $\theta = (p, \mu, \Sigma)$, to compute the minimal representative of the class $[\theta]$ of equivalent tuples of parameters using the following proposition:

**Proposition 4.14.** *Let $\theta = (p, \mu, \Sigma) \in \Theta$ be parameters. Then, the minimal representative of the class $[\theta]$ as defined at Definition 4.13 is $\theta^{min} = (p^{min}, \mu^{min}, \Sigma^{min})$ where*

$$p^{min} = e^{\frac{1}{d} \sum_{i=1}^d \mu_i} p, \quad \mu^{min} = \mu - \frac{1}{d} \sum_{i=1}^d \mu_i \nu, \quad \Sigma^{min} = \Sigma$$

*where we recall that $\nu = (1, \cdots, 1, 0, \cdots, 0) \in \mathbb{R}^{d(d+1)/2}$.*

This minimal representative will be used in the following.

## 5. Estimation of the parameters of a wrapped Gaussian distribution

In this section, we tackle the parameter estimation problem of a wrapped Gaussian given samples. After introducing the Maximum Likelihood Estimator we will use, we lead some synthetic experiments to assess its performance.

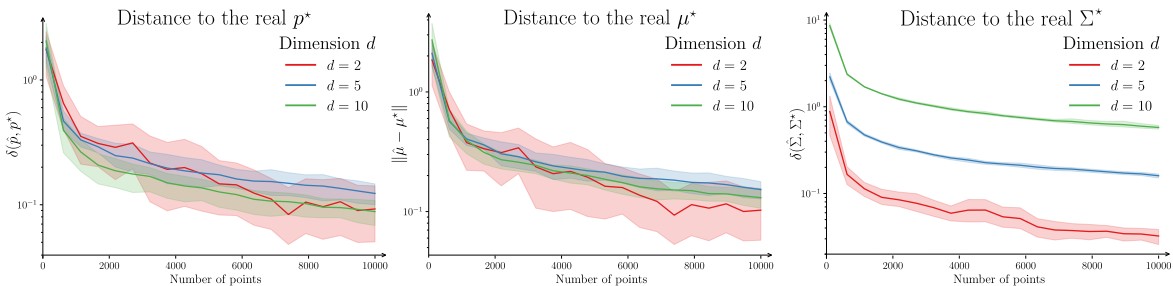

*Figure 3.* Results of the synthetic experiment on the estimation of parameters of a wrapped Gaussian using an MLE.

## 5.1. A Maximum Likelihood Estimator

Let $x_1, ..., x_N$ be $N$ SPD matrices of size $d \times d$ independently sampled from a wrapped Gaussian with unknown parameters $\theta^\star = (p^\star, \mu^\star, \Sigma^\star)$ supposed to be minimal in the sens of Definition 4.13. Our goal is to estimate $\theta^\star$ given the samples $(x_1, ..., x_N)$ using a *Maximum Likelihood Estimator* (MLE) (see Section 9.3 of Wasserman 2004). Let us introduce the likelihood $\mathcal{L}_N$ of the model:

$$\mathcal{L}_N(p; \mu, \Sigma) = \prod_{i=1}^{N} f_{p;\mu,\Sigma}(x_i)$$

where $f_{p;\mu,\Sigma}$ is the density of the wrapped Gaussian $WG(p; \mu, \Sigma)$ as given in Equation (6). We will also consider the log-likelihood $\ell_N$ defined as $\ell_N(p; \mu, \Sigma) = \log \mathcal{L}_N(p; \mu, \Sigma)$. Then, we define the classical MLE $\hat{\theta}_N = (\hat{p}_N, \hat{\mu}_N, \hat{\Sigma}_N)$ as the parameter $\theta$ that maximizes $\mathcal{L}_N$ (or equivalently $\ell_N$). In the Euclidean setting, one has a closed form of the MLE of $\mu$ and $\Sigma$, obtained by computing the gradient of $\ell_N$. When dealing with wrapped Gaussians on $\mathcal{P}_d$, we were not able to derive a closed form for the MLE $\hat{p}_N$ of the parameter $p$. Moreover, it is unlikely that there exists such a closed form, as for example, there is no closed form for the Riemannian mean on $\mathcal{P}_d$ (Moakher, 2005). Nevertheless, the MLE of $\mu$ and $\Sigma$ are analogous to the Euclidean setting but depend on $p^\star$:

**Proposition 5.1.** *The MLE $\hat{\mu}_N$ and $\hat{\Sigma}_N$ of the parameters $\mu$ and $\Sigma$ of the wrapped Gaussian are:*

$$\hat{\mu}_N = \frac{1}{N} \sum_{i=1}^{N} \mathrm{VLog}_{p^\star}(x_i),$$

$$\hat{\Sigma}_N = \frac{1}{N} \sum_{i=1}^{N} \left(\mathrm{VLog}_{p^\star}(x_i) - \hat{\mu}_N\right) \left(\mathrm{VLog}_{p^\star}(x_i) - \hat{\mu}_N\right)^\top$$

*where $\mathrm{VLog}_{p^\star} = \mathrm{Vect}_{p^\star} \circ \mathrm{Log}_{p^\star}$.*

One can note that in proposition 4.7 of Galaz-Garcia et al. (2022), they also have a closed from for the MLE of $\Sigma$ that depends on $p^\star$ without any closed form for $\hat{p}_N$. In practice, we used a Riemannian Conju-

gate Gradient algorithm (Boumal, 2023) on the product manifold $\Theta = \mathcal{P}_d \times \mathbb{R}^{d(d+1)/2} \times \mathcal{P}_{d(d+1)/2}$ to compute the optimal parameters $(\hat{p}_N, \hat{\mu}_N, \hat{\Sigma}_N)$. We implemented this MLE in Python using the toolbox Pymanopt (Townsend et al., 2016). The codes for the different experiments is available at https://github.com/thibaultdesurrel/wrapped_gaussians_SPD.

This estimation problem can become challenging as the dimension $d$ of the considered SPD matrices increases. In fact, the number of coefficients to estimate is:

$$\underbrace{\frac{d(d+1)}{2}}_{p} + \underbrace{\frac{d(d+1)}{2}}_{\mu} + \underbrace{\frac{d^2(d+1)^2 + 2d(d+1)}{8}}_{\Sigma}$$

which grows at a rate of $O(d^4)$. For example, if $d = 10$, there are $1,650$ coefficients to estimate and if $d = 30$, the number jumps to $109,275$. One should thus make sure that the number of samples $N$ is sufficiently big for results of the MLE to make sense. To get around this issue when the number of samples is small, one can assume that the covariance matrix $\Sigma$ is diagonal, which reduces the number of coefficients to $O(d^2)$. This assumption implies independent entries of the SPD matrices and will be used in Section 6.4.

In other works, such as in Chevallier et al. (2022) or in Chevallier & Guigui (2020), the authors use the method of moments (see Section 9.2 of Wasserman 2004) to estimate the parameter. In our case, we can use the method of moment only when we know *a priori* that $\mu^\star = 0$. Then, as given in Proposition 4.8, a mean of the wrapped Gaussian is $p^\star$, therefore, it can be estimated using the Riemannian mean $\hat{p}_N = \mathfrak{G}(x_1, ..., x_N)$ (Moakher, 2005), and $\Sigma$ can be estimated using Proposition 5.1. However, in a more general case of $\mu^\star \neq 0$, estimating $p^\star$ using the Riemannian mean does not lead to the correct estimation of the true parameters. We give more details on why in Appendix H.

## 5.2. Synthetic experiments

We led some synthetic experiments to evaluate the MLE's performances. For this, we sampled $N$ points from a wrapped Gaussian in $\mathcal{P}_d$ whose minimal parameters

$(p^\star, \mu^\star, \Sigma^\star)$ are known. Here, $\Sigma^\star$ is always chosen to be a full SPD matrix. Then, we optimize the MLE to find $(\hat{p}_N, \hat{\mu}_N, \hat{\Sigma}_N)$, and compare the true and minimal estimated parameters: for $p$ and $\Sigma$, we compare them using the AIRM distance and for $\mu$, we use $\| \cdot \|_2$. In this experiment, we looked at how the estimation error evolves when the number of samples $N$ grows from 100 to 10,000. We also compare different dimensions $d \in \{2, 5, 10\}$. More details on the experimental setup are given in Appendix I. The results of this experiment can be found in Figure 3. We can see that, as one would expect, as the number $N$ of points sampled grows, the estimation error decreases. Moreover, we can remark that the dimension $d$ does not affect the results of the estimation of $p$ and $\mu$, but really affects the estimation of $\Sigma$. The higher the dimension $d$, the higher the error $\delta(\hat{\Sigma}_N, \Sigma^\star)$ is. This is coherent as the number of parameters of $\Sigma$ grows as $O(d^4)$, so even a small increase of the dimension $d$ leads to an important increment in the estimation error of $\Sigma$. We also led some experiments in the case where the covariance matrix $\Sigma$ is diagonal. We show, in Appendix J, that in this case, one needs fewer samples to have a good estimation of $\Sigma$ when the dimension $d$ rises.

# 6. Classification using wrapped Gaussians

In this section, we demonstrate that widely used geometry-aware classifiers on $\mathcal{P}_d$ can be integrated into a probabilistic framework. We also introduce new classifiers based on the wrapped Gaussian and conduct experiments on real-world data from various applications.

## 6.1. Classifiers used for SPD matrices

**MDM**  The *Minimum Distance to Mean* (MDM) algorithm described in Barachant et al. (2010) is a popular classifier for SPD matrices. Given a training set of labeled SPD matrices, the MDM computes the Riemannian mean (Moakher, 2005) $\mathfrak{G}_k$ of each class $k \in \{1, ..., K\}$. Then, given a new SPD matrix $p$, the predicted class $\hat{k}$ is the class for which the distance between $p$ and $\mathfrak{G}_k$ is the smallest.

**LDA and QDA**  In a Euclidean setting, the *Linear Discriminant Analysis* (LDA) (section 4.3 of Hastie et al. 2009) is a classifier that assumes that each class $k \in \{1, ..., K\}$ is modeled by a multivariate Gaussian $\mathcal{N}(\mu_k, \Sigma)$ where the covariance matrix $\Sigma$ is shared among all the classes. First, the parameters of each class are learned using an MLE on the training data. Then, to classify a new point $z$, LDA compares the log-likelihood of $z$ according to each class and chooses the class that has the highest log-likelihood. As the covariance matrix is shared among the classes, the decision boundaries are linear, which led to its name: *linear discriminant analysis*. When one assumes that the covariance matrices are not equal among the classes, i.e. each

class is modeled by $\mathcal{N}(\mu_k, \Sigma_k)$, the decision boundaries are quadratic. This classifier is called *Quadratic Discriminant Analysis* (QDA). One can restrict the covariance matrices to be diagonal, which leads to the *Diagonal LDA* and *Diagonal QDA* classifiers (Dudoit et al., 2002). Then, the Diagonal QDA classifier is equivalent to the Gaussian Naive Bayes classifier (see chapter 8 sec 3.3 of Bishop 2007).

A possible extension of LDA (or QDA) to the manifold of SPD matrices $\mathcal{P}_d$ is call *Tangent Space LDA* (or *Tangent Space QDA*) and is described in part IV B. of Barachant et al. (2012). The Riemannian mean $\mathfrak{G}$ of the training set is computed, and all training points are sent to the tangent space $T_{\mathfrak{G}} \mathcal{P}_d$ *via* the logarithm map $\mathrm{Log}_{\mathfrak{G}}$. Then, a classical LDA (or QDA) can be used in this Euclidean space.

**Other classifiers from SPD matrices**  Other classifiers that have been developed for SPD matrices. For example, Multinomial Logistics Regression has been extended for SPD matrices in Chen et al. (2024). They rely on metric that are pulled back from the Euclidean space which is not the case of the AIRM metric we use in our work. Several deep learning approaches have been proposed to classify SPD matrices (Huang & Van Gool, 2017; Brooks et al., 2019; Nguyen, 2021). However, most of these approaches distort the geometry of the manifold and are out of the scope of this work, as our approach does not rely on deep learning.

## 6.2. A general probabilistic framework

Our goal is to show that the MDM, Tangent Space LDA and Tangent Space QDA can be seen as part of a probabilistic framework on the manifold of SPD matrices. More precisely, we will show that the previous classifiers can be rewritten as Maximum Likelihood based classifiers (like the classical LDA or QDA) where the different classes are modeled using distributions on the manifold $\mathcal{P}_d$. Let us consider $K$ classes of labeled SPD matrices and let us denote $\alpha_k$ the modeled distribution of class $k$.

**MDM**  For the MDM classifier, we first need to recall the isotropic Gaussians on $\mathcal{P}_d$ introduced in Said et al. (2018). Let $\bar{y} \in \mathcal{P}_d$ and $\sigma > 0$, then, the isotropic Gaussian denoted $G(\bar{y}, \sigma)$ is defined by the following density:

$$\forall y \in \mathcal{P}_d, \ f_{\bar{y},\sigma}(y) = \frac{1}{\zeta(\sigma)} \exp\left[ -\frac{\delta(y, \bar{y})^2}{2\sigma^2} \right]$$

where $\zeta(\sigma)$ is a normalizing constant. This normalizing constant depends only on $\sigma$, and not on $\bar{y}$ as shown in proposition 1 of Said et al. (2018). If one supposes that each class is modeled by an isotropic Gaussian with a shared $\sigma$ among all the classes i.e. $\alpha_k = G(\bar{y}_k, \sigma)$ then, the MDM is equivalent to a maximum likelihood classifier. Here, the spread $\sigma$ of the isotropic Gaussians does not play any role

| Dataset | Application | Dimension | # matrices | # classes | Reference |
|---|---|---|---|---|---|
| BNCI2014004 | BCI | $3 \times 3$ | 720 x 9 subjects | 2 | (Leeb et al., 2007) |
| Zhou2016 | BCI | $5 \times 5$ | 320 x 4 subjects | 2 | (Zhou et al., 2016) |
| AirQuality | Atmospheric data | $6 \times 6$ | 102 | 3 | (Smith et al., 2022) |
| Indiana Pines | Hyperspectral imaging | $5 \times 5$ | $14,641$ | 12 | (Baumgardner et al., 2015) |
| Pavia Univ. | Hyperspectral imaging | $5 \times 5$ | $185,176$ | 6 | - |
| Salinas | Hyperspectral imaging | $5 \times 5$ | $94,184$ | 17 | - |
| Textile | Image Analysis | $10 \times 10$ | $16,000$ | 2 | (Bergmann et al., 2021) |
| BreizhCrops | Multispectral imaging | $13 \times 13$ | $177,658$ | 6 | (Rußwurm et al., 2020) |

*Table 1.* Summary of the datasets used for the experiments.

| Dataset | Acc. MDM | Acc. TS-LDA | Acc. TS-QDA | Acc. Ho-WDA | Acc. He-WDA |
|---|---|---|---|---|---|
| BNCI2014004 | 78.71 $(\pm 14.53)$ | **78.73** $(\pm 14.52)$ | 76.07 $(\pm 13.94)$ | 75.38 $(\pm 14.40)$ | 74.37 $(\pm 14.73)$ |
| Zhou2016 | 91.18 $(\pm 5.51)$ | **91.21** $(\pm 5.50)$ | 89.45 $(\pm 7.43)$ | 85.92 $(\pm 9.20)$ | 82.86 $(\pm 11.65)$ |
| Air Quality | 94.05 $(\pm 6.53)$ | 94.05 $(\pm 6.53)$ | **97.05** $(\pm 4.45)$ | 96.05 $(\pm 4.17)$ | **97.00** $(\pm 4.47)$ |
| Indiana | 58.01 $(\pm 0.72)$ | 67.07 $(\pm 0.53)$ | 73.38 $(\pm 0.45)$ | 73.74 $(\pm 0.51)$ | **74.30** $(\pm 0.83)$ |
| Pavia Uni. | 72.32 $(\pm 0.59)$ | 84.61 $(\pm 0.06)$ | 87.16 $(\pm 0.09)$ | **87.36** $(\pm 0.07)$ | 85.54 $(\pm 0.15)$ |
| Salinas | 36.42 $(\pm 0.12)$ | 46.30 $(\pm 0.17)$ | 69.87 $(\pm 0.21)$ | **71.20** $(\pm 0.33)$ | 62.39 $(\pm 0.22)$ |
| Textile | 83.08 $(\pm 0.62)$ | 83.12 $(\pm 0.63)$ | 86.03 $(\pm 0.66)$ | **86.26** $(\pm 0.59)$ | 85.93 $(\pm 0.77)$ |
| BreizhCrops | 45.48 $(\pm 0.25)$ | 47.67 $(\pm 0.32)$ | 50.72 $(\pm 0.28)$ | **54.66** $(\pm 0.45)$ | 51.33 $(\pm 0.58)$ |

*Table 2.* Accuracy of the different classifiers on the different datasets we consider.

in the classification process. So, during training, one only has to estimate $\bar{y}_k$ for each class, which is the center of mass and can be estimated using the Riemannian mean (see proposition 7 of Said et al. 2018).

**Tangent Space LDA or QDA** For the Tangent Space LDA, we will leverage the Wrapped Gaussians introduced in Section 4. Suppose that each class is modeled by a wrapped Gaussian centered at $\mathfrak{G}$, the Riemannian mean of the training set, and with a shared covariance matrix $\Sigma$ for the Tangent Space LDA i.e. $\alpha_k = \mathrm{WG}(\mathfrak{G}; \mu_k, \Sigma)$ or with one covariance matrix $\Sigma_k$ per class for the Tangent Space QDA i.e. $\alpha_k = \mathrm{WG}(\mathfrak{G}; \mu_k, \Sigma_k)$. Then, the Tangent Space LDA (or Tangent Space QDA) is a maximum likelihood classifier based on those distributions.

### 6.3. Wrapped Discriminant Analysis

Having placed the various classifiers that are used on the manifold of SPD matrices in a probabilistic framework, we propose a new maximum likelihood classifier based on the wrapped Gaussians introduced in Section 4. First, let us model each class by a wrapped Gaussian with a shared covariance matrix $\Sigma$ among the classes: $\alpha_k = \mathrm{WG}(p_k; \mu_k, \Sigma)$. To learn the parameters of each class, we optimize an MLE on the whole model to find the parameters:

$$(p_1, ..., p_K, \mu_1, ..., \mu_K, \Sigma) \in \operatorname*{argmax}_{\boldsymbol{p}, \boldsymbol{\mu}, \Sigma} \prod_{k=1}^{K} \prod_{i=1}^{N_k} f_{p_k; \mu_k, \Sigma}(x_i^k)$$

where $(x_i^k)_{i=1,..,N_k}$ are the training points of class $k$. The

implementation is the same as in Section 5. We call it the *Homogeneous Wrapped Discriminant Analysis* (Ho-WDA).

As for the QDA, we propose another version of this classifier where each class has its own covariance matrix $\Sigma_k$: $\alpha_k = \mathrm{WG}(p_k; \mu_k, \Sigma_k)$. We call this classifier the *Heterogeneous Wrapped Discriminant Analysis* (He-WDA). In that case, an MLE is optimized on each class individually, as in Section 5:

$$\forall k \in \{1, ..., K\}, \ (p_k, \mu_k, \Sigma_k) \in \operatorname*{argmax}_{p, \mu, \Sigma} \prod_{i=1}^{N_k} f_{p; \mu, \Sigma}(x_i^k).$$

### 6.4. Experiments

In this section, we want to compare the Ho-WDA and He-WDA to the other classifiers (MDM, Tangent Space LDA denoted TS-LDA and Tangent Space QDA denoted TS-QDA) used on the manifold of SPD matrices $\mathcal{P}_d$ and detailed in Section 6.1. For this, we lead some experiments on 8 different real datasets coming from several applications. We give a summary of the datasets used in Table 1 and more details on each one of them in Appendix K. For this experiment, we restricted ourselves to the case where the covariance matrices $\Sigma$ are diagonal. Therefore, one has fewer coefficients to estimate and the MLE needs less points to converge. We give the accuracy of the classifiers we study on the different datasets at Table 2.

We can see two different behaviors. First, on the datasets with a lot of matrices (Textile, Salinas, Indiana Pines, Pavia Univ., BreizhCrops), the Ho-WDA and He-WDA perform the best. The number of parameters to estimate is high, so the more samples one has, the better the estimation will be

as we illustrated in the synthetic experiments at Section 5. For BreizhCrops, even if the matrices are of size $13 \times 13$, we have a lot of points $(177, 658)$ so the MLE is able to correctly estimate all the parameters. Secondly, on the BCI datasets (BNCI2014004 and Zhou2016), we have significantly less points (less than $1, 000$) so the estimation of the parameters of the underlying wrapped Gaussians is less precise. In this case, one can see that the MDM and the TS-LDA perform the best and the Ho-WDA and He-WDA perform less well. However, on the AirQuality dataset, the Ho-WDA and He-WDA perform the best. This is interesting as the number of matrices available in the dataset is small (102). An explanation could be that the underlying distribution of the data is not very complex, so a few points are enough to correctly estimate them. In BCI datasets, the distribution is more complex, so one needs more points to correctly estimate the distribution. Finally, we do not observe a clear dominance of the He-WDA over the Ho-WDA. This is similar to the difference between the LDA and QDA where, often, an LDA can correctly classify data. In theory, the Ho-WDA should be a special case of the He-WDA where the covariance matrices for each class are the same. However, in practice, as the number of points per class can be small and as the number of parameters to estimate is higher for the He-WDA, the estimation of the covariance matrices per class can be noisy and the performance of the He-WDA can be worse than that of the Ho-WDA. For example, for the dataset Salinas, some classes have only a few hundred samples, which is not enough to estimate the covariance matrix accurately. The Ho-WDA, on the other hand, estimates a single covariance matrix for all the classes and is less sensitive to the number of samples per class.

## 7. Conclusion

In this work, we present a generalization of non-isotropic multivariate Gaussians on the manifold of SPD matrices: *Wrapped Gaussians*, and we give some theoretical properties. We solved the non-identifiability of our model by defining an equivalence relation between the set of parameters that define the same wrapped Gaussian. We also give all the tools needed to use the distribution in practice, such as an easy-to-use sampling algorithm or an MLE that correctly estimates the parameters of a wrapped Gaussian. Finally, we showed that the MDM, TS-LDA and TS-QDA classifiers can be seen as part of a probabilistic framework using wrapped Gaussians. We introduced two new classifiers based on the wrapped Gaussian: Ho-WDA and He-WDA. We showed that the Ho-WDA and He-WDA perform well on real data when the number of samples is sufficient. In future work, we plan to investigate the use of wrapped Gaussians to perform data augmentation or transfer learning. Moreover, as we have developed in the paper a geometry-aware Gaussian distribution, it becomes possible to extend all the

classical machine learning models that rely on Gaussian distributions to the manifold of SPD matrices. We are aware that computing exponential and logarithmic maps remains a bottleneck in SPD matrix geometry. However, a trade-off may exist between computational cost and performance gains. Theoretically, Euclidean Gaussian-based methods extend to $\mathcal{P}_d$ *via* our wrapped Gaussian, though practical challenges will arise in applications, requiring careful choices. For example, one could develop Gaussian Mixture Models, Hidden Markov Models or Variational Autoencoders on $\mathcal{P}_d$ using wrapped Gaussians. These models could then be applied to various tasks such as clustering, sequence modeling, or anomaly detection on manifold-valued data. It could also be possible to explore the use of wrapped Gaussians in the context of Bayesian inference, which could open new avenues for probabilistic modeling and uncertainty quantification in manifold-based data analysis.

## Acknowledgments

This work was funded by the French National Research Agency for project PROTEUS (grant ANR-22-CE33-0015-01). Part of this work was carried out while Florian Yger was a member of PSL-Dauphine University, and he acknowledges the support of the ANR as part of the "Investissements d'avenir" program, reference ANR-19-P3IA-0001 (PRAIRIE 3IA Institute). Sylvain Chevallier is supported by DATAIA (ANR-17-CONV-0003). The authors would also like to thank Bastien Cavarretta that helped inspire Theorem 4.7 and Lucas Gnecco Heredia for his help in proofreading the paper.

## Impact statement

This work introduces a flexible and theoretically grounded extension of Gaussian distributions to the Riemannian manifold of Symmetric Positive Definite (SPD) matrices, which appear in numerous application domains such as neuroimaging (e.g., EEG/BCI), remote sensing, atmospheric modeling, and computer vision. By leveraging a non-isotropic wrapped Gaussian model, our approach respects the intrinsic geometry of SPD matrices, offering a more principled statistical modeling framework for manifold-valued data.

The proposed distribution, along with the associated maximum likelihood estimator and probabilistic classifiers, can enhance the interpretability, robustness, and effectiveness of machine learning models in applications where geometric constraints are crucial. For example, in Brain-Computer Interfaces (BCI), this work may contribute to more accurate and stable classification of neural signals, potentially benefiting assistive technologies. In environmental sciences, the method can aid in more accurate statistical modeling of air quality data represented as covariance matrices.

However, as with any advancement in data modeling, especially in sensitive domains such as neuroscience, there is a possibility that improved interpretability or classification performance could be misused. For instance, fine-grained neural decoding could be exploited for persuasive technologies or behavioral profiling, raising ethical concerns around privacy and consent. We thus emphasize the importance of applying these techniques within responsible and ethically guided frameworks.

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

# A. The vectorization used

In Section 3.2, we defined an isomorphism between the tangent space $T_p\mathcal{P}_d$ at point $p \in \mathcal{P}_d$ and $\mathbb{R}^{d(d+1)/2}$. This isomorphism is called the vectorization and is denoted $\text{Vect}_p$. This isomorphism is not the only one that exists between the two spaces, so in this section we will motivate our choice of the vectorization and give some of its properties.

Let $p \in \mathcal{P}_d$. We recall that the tangent space $T_p\mathcal{P}_d$ at point $p$ is a Euclidean space once equipped with the inner product defined at Equation (1). Therefore, one can unveil an orthonormal basis of $T_p\mathcal{P}_d$. One can see that the tangent space at the identity $T_{I_d}\mathcal{P}_d$ is the classical Euclidean space $\mathcal{S}_d$ equipped with the Frobenius inner product. Therefore, one can easily build an orthonormal basis of $T_{I_d}\mathcal{P}_d$ and then, transport it to the other tangent spaces.

**Proposition A.1** (Orthonormal basis of the tangent spaces). *Let $e_{ij}$ be the $d \times d$ matrix with a 1 at position $(i, j)$ and zeros everywhere else. Then*

- *An orthonormal basis of $(T_{I_d}\mathcal{P}_d, \langle \cdot, \cdot \rangle_{I_d})$ is $(E_{I_d,ij})_{i \leq j}$ defined as follows:*

$$E_{I_d,ij} = \begin{cases} \frac{1}{\sqrt{2}}(e_{ij} + e_{ji}) & \text{for } i < j, \\ e_{ii} & \text{for } i = j. \end{cases}$$

- *An orthonormal basis of $(T_p\mathcal{P}_d, \langle \cdot, \cdot \rangle_p)$ is $(E_{p,ij})_{i \leq j}$ where $E_{p,ij} = p^{1/2}E_{I_d,ij}p^{1/2}$.*

*Proof.* One has that $T_{I_d}\mathcal{P}_d \simeq \mathcal{S}_d$ and that $\langle \cdot, \cdot \rangle_{I_d}$ is the Frobenius inner product, so one can use the classical basis of $\mathcal{S}_d$ to build an orthonormal basis of $T_{I_d}\mathcal{P}_d$. Then, by transporting the basis of $T_{I_d}\mathcal{P}_d$ to $T_p\mathcal{P}_d$ using the isometry $x \mapsto p^{1/2}xp^{1/2}$, one has a basis of $T_p\mathcal{P}_d$. It is still orthonormal as $x \mapsto p^{1/2}xp^{1/2}$ is an isometry. $\qquad\square$

Let us give another intuition on the choice of this basis for $(T_p\mathcal{P}_d, \langle \cdot, \cdot \rangle_p)$:

**Proposition A.2.** *The basis $(E_{p,ij})_{i \leq j}$ of $(T_p\mathcal{P}_d, \langle \cdot, \cdot \rangle_p)$ given at Proposition A.1 is the parallel transport of the basis $(E_{I_d,ij})_{i \leq j}$ of $(T_{I_d}\mathcal{P}_d, \langle \cdot, \cdot \rangle_{I_d})$ from $T_{I_d}\mathcal{P}_d$ to $T_p\mathcal{P}_d$.*

*Proof.* According to Equation 22 of Sra & Hosseini (2015), in the case of $\mathcal{P}_d$, the parallel transport $\Gamma_{I_d \to p}$ from $T_{I_d}\mathcal{P}_d$ to $T_p\mathcal{P}_d$ is:

$$\forall u \in T_{I_d}\mathcal{P}_d, \ \Gamma_{I_d \to p}(u) = p^{1/2}up^{1/2}.$$

The result follows from the definition of $(E_{p,ij})_{i \leq j}$. $\qquad\square$

Now that we have an orthonormal basis of the tangent space $T_p\mathcal{P}_d$, we give the link between this basis and the vectorization $\text{Vect}_p$:

**Proposition A.3.** *Let $(E_{p,ij})_{i \leq j}$ be the orthonormal basis of the tangent space $T_p\mathcal{P}_d$ described at Proposition A.1. Let $u \in T_p\mathcal{P}_d$. Then,*

$$\text{Vect}_p(u) = (\langle u, E_{p,11} \rangle_p, \langle u, E_{p,12} \rangle_p, \langle u, E_{p,22} \rangle_p, \cdots, \langle u, E_{p,d-1d} \rangle_p, \langle u, E_{p,dd} \rangle_p).$$

*Proof.* We start with the case where $p = I_d$. Let $u = [[u_{ij}]] \in T_{I_d}\mathcal{P}_d \simeq \mathcal{S}_d$. We simply need to show that, for $i \leq j$, one has

$$\langle u, E_{I_d,ij} \rangle_{I_d} = \begin{cases} u_{ii} & \text{if } i = j, \\ \sqrt{2}u_{ij} & \text{if } i < j. \end{cases}$$

One has, when $i = j$:

$$\langle u, E_{I_d,ii} \rangle_{I_d} = \langle u, e_{ii} \rangle_{I_d} = \text{tr}(ue_{ii}) = u_{ii}.$$

And when $i < j$:

$$\langle u, E_{I_d,ij} \rangle_{I_d} = \langle u, \frac{1}{\sqrt{2}}(e_{ij} + e_{ji}) \rangle_{I_d} = \frac{1}{\sqrt{2}}(\text{tr}(ue_{ij}) + \text{tr}(ue_{ji})) = \sqrt{2}u_{ij}.$$

Therefore, one has the results for $\text{Vect}_{I_d}$.

Now, in the general case of $p \in \mathcal{P}_d$, one has that, for $i \leq j$

$$\langle u, E_{p,ij} \rangle_p = \langle u, p^{1/2} E_{I_d,ij} p^{1/2} \rangle_p = \langle p^{-1/2} u p^{-1/2}, E_{I_d,ij} \rangle_{I_d}.$$

By using the definition of $\mathrm{Vect}_p(u) = \mathrm{Vect}_{I_n}(p^{-1/2} u p^{-1/2})$ and the result for $p = I_d$, one has the result. $\square$

The previous proposition helps us motivate the choice of the vectorization. Indeed, this vectorization is simply the coordinates of the tangent vector $u$ in the orthonormal basis $(E_{p,ij})_{i \leq j}$ of the tangent space $T_p \mathcal{P}_d$. Now that we are more convinced on this choice of vectorization, we give some of its important properties.

**Proposition A.4.** • *Let $u \in T_p \mathcal{P}_d$, then*

$$\| \mathrm{Vect}_p(u) \|_2^2 = \mathrm{Vect}_p(u)^\top \mathrm{Vect}_p(u) = \| u \|_p^2 := \sqrt{\langle u, u \rangle_p}.$$

*Therefore, $\mathrm{Vect}_p$ is not only an isomorphism, it is an isometry between $(T_p \mathcal{P}_d, \langle \cdot, \cdot \rangle_p)$ and $(\mathbb{R}^{d(d+1)/2}, \| \cdot \|_2)$.*

• *Let $u \in T_p \mathcal{P}_d$, then*

$$\| \mathrm{Vect}_p(\mathrm{Log}_p u) \|_2^2 = \mathrm{Vect}_p(\mathrm{Log}_p u)^\top \mathrm{Vect}_p(\mathrm{Log}_p u) = \delta(p, u)^2.$$

*Proof.* Let us prove the two points of the proposition.

• Let $u \in T_p \mathcal{P}_d$. One has, using Proposition A.3,

$$\| \mathrm{Vect}_p(u) \|_2^2 = \mathrm{Vect}_p(u)^\top \mathrm{Vect}_p(u) = \sum_{i \leq j} \langle u, E_{p,ij} \rangle_p^2.$$

As $(E_{p,ij})_{i \leq j}$ is an orthonormal basis of the Euclidean space $(T_p \mathcal{P}_d, \langle \cdot, \cdot \rangle_p)$, one has that

$$\sum_{i \leq j} \langle u, E_{p,ij} \rangle_p^2 = \| x \|_p.$$

which proves the first point.

• Let $u \in T_p \mathcal{P}_d$. One has, using the previous point, the definition of $\| \cdot \|_p$, the expression of the Riemannian logarithm given at Equation (3) and the expression of the AIRM distance given at Equation (2):

$$\| \mathrm{Vect}_p(\mathrm{Log}_p u) \|_2^2 = \| \mathrm{Log}_p u \|_p^2 = \| p^{-1/2} \mathrm{Log}_p u \, p^{-1/2} \|_F = \| \log(p^{-1/2} u p^{-1/2}) \|_F = \delta(p, u)^2.$$

$\square$

Finally, let us give a consequence of the previous property on the Jacobian of the vectorization: as $\mathrm{Vect}_p$ is an isometry, there is no volume change *via* the vectorization.

# B. The push-forward

In this section, we give the definition and an important result on the push-forward measure. One can find more information on the push-forward measure in Section 3.6 of Bogachev (2007).

**Definition B.1** (Pushforward measure). Given two measurable spaces $(\mathcal{X}, \Omega_{\mathcal{X}})$ and $(\mathcal{Y}, \Omega_{\mathcal{Y}})$, a measurable map $f \colon \Omega_{\mathcal{X}} \to \Omega_{\mathcal{Y}}$ and a measure $\mu \colon \Omega_{\mathcal{X}} \to [0, +\infty]$, the *pushforward* of $\mu$ is defined to be the measure $f \# \mu \colon \Omega_{\mathcal{Y}} \to [0, +\infty]$ given by

$$\forall B \in \Omega_{\mathcal{Y}}, \ (f \# \mu)(B) = \mu(f^{-1}[B]).$$

where $f^{-1}[B]$ is the preimage of $B$ by $f$.

We now give the most important result on pushforward measures: the change of variables

**Theorem B.2** (Change of variables). *Let $\mu$ be a non-negative measure. An $\Omega_{\mathcal{Y}}$-measurable function $g$ on $\mathcal{Y}$ is integrable with respect to the pushforward measure $f \# \mu$ if and only if the function $g \circ f$ is integrable with respect to the measure $\mu$. In this case, one has:*

$$\int_{\mathcal{Y}} g(y) \mathrm{d}(f \# \mu)(y) = \int_{\mathcal{X}} (g \circ f)(x) \mathrm{d}\mu(x).$$

## C. The Jacobian determinant of the exponential map on the manifold of SPD matrices

In this section, we give a proof of the result stated at Proposition 4.3. More specifically, we want to show that the Jacobian determinant of the exponential map $\mathrm{Exp}_{I_d}$ at identity is:

$$J_{I_d}(u) = 2^{d(d-1)/2} \prod_{i<j} \frac{\sinh\left(\frac{\lambda_i(u)-\lambda_j(u)}{2}\right)}{\lambda_i(u)-\lambda_j(u)} \tag{7}$$

where the $(\lambda_i(u))_i$ are the eigenvalues of $u$.

Let us start by recalling that, when the tangent plan of interest is at the identity $I_d$, the Riemannian exponential map $\mathrm{Exp}_{I_d}$ is simply the matrix exponential $\exp$:

$$\mathrm{Exp}_{I_d} \colon u \in T_{I_d}\mathcal{P}_d \mapsto \exp(u) \in \mathcal{P}_d.$$

One can see this result using the expression of the Riemannian exponential given in Equation (3).

To prove the relation Equation (7), we will start by the case where $u$ is a diagonal matrix. We will then extend the result to the general case.

**Case 1: $u \in T_{I_d}\mathcal{P}_d$ diagonal**  Let us consider $u \in T_{I_d}\mathcal{P}_d$ diagonal, $u = \mathrm{diag}(\lambda_1, ..., \lambda_d)$. In the following, we will denote by $\Psi$ the differential of the Riemannian exponential in $u$: $\Psi = \mathrm{d}\,\mathrm{Exp}_{I_d}(u) = \mathrm{d}\exp(u)$. We therefore want to compute the determinant of $\Psi$: $\det\Psi$. One has that $\Psi \colon T_{I_d}\mathcal{P}_d \to T_{\exp(u)}\mathcal{P}_d$, where we have identified $T_u T_{I_d}\mathcal{P}_d$ with $T_{I_d}\mathcal{P}_d$. To compute the determinant, one need to choose adequate bases in both tangent spaces $T_{I_d}\mathcal{P}_d$ and $T_{\exp(u)}\mathcal{P}_d$. By "adequate", we mean that the transformation between the two bases does not imply any volume change. For this, we consider for $T_{I_d}\mathcal{P}_d$ the basis $(E_{I_d,ij})_{i\leq j}$ and for $T_{\exp(u)}\mathcal{P}_d$ the basis $(E_{\exp(u),ij})_{i\leq j}$ as defined at Proposition A.1. According to Proposition A.2, the transformation from the first to the second basis is the parallel transport, which does not imply any volume changes, as the parallel transport is an isometry (see Prop 10.36 of Boumal 2023).

Now that we have our two basis, we want to compute the matrix of $\Psi$ in those two bases. For this, we need to compute $\Psi(E_{I_d,ij})$ and express it in the basis $(E_{\exp(u),ij})_{i\leq j}$. As $u$ is diagonal, we can use the *Daletskii-Krein formula* (see Daletskii & Krein (1965) or Equation 2.40 of Bhatia 2007) that states the following in our case: for $h \in T_p\mathcal{P}_d$,

$$\Psi(h) = \left[\left[\exp^{[1]}(u)_{ij}h_{ij}\right]\right] \tag{8}$$

where

$$\exp^{[1]}(u)_{ij} = \begin{cases} e^{\lambda_i} & \text{for } i = j, \\ \frac{e^{\lambda_i}-e^{\lambda_j}}{\lambda_i-\lambda_j} & \text{for } i \neq j. \end{cases}$$

Using the previous formula, one can compute $\Psi(E_{I_d,ij})$ for $i \leq j$. Now, one needs to compute the coefficients of $\Psi(E_{I_d,ij})$ in the basis $(E_{\exp(u),kl})_{k\leq l}$. As the basis $(E_{\exp(u),kl})_{k\leq l}$ is orthonormal, one simply needs to compute the dot product between $\Psi(E_{I_d,ij})$ and one element of the basis to get the corresponding coefficient. For $k \leq l$, one has:

$$
\begin{aligned}
\langle \Psi(E_{I_d,ij}), E_{\exp(u),kl}\rangle_{\exp(u)} &= \langle \Psi(E_{I_d,ij}), \exp(u/2)E_{I_d,kl}\exp(u/2)\rangle_{\exp(u)} && \text{using the definition of } E_{\exp(u),kl} \\
&= \mathrm{tr}\left(\Psi(E_{I_d,ij})\exp(-u/2)E_{I_d,kl}\exp(-u/2)\right) && \text{using the definition of the AIRM metric} \\
&= \langle \exp(-u/2)\Psi(E_{I_d,ij})\exp(-u/2), E_{I_d,kl}\rangle_{I_d}
\end{aligned}
$$

Therefore, it is the coefficient $(k, l)$ of the matrix $\exp(-u/2)\Psi(E_{I_d,ij})\exp(-u/2)$ (up to a factor $\sqrt{2}$ when $k \neq l$). We now need to compute this matrix. As $u$ is diagonal, $u = \mathrm{diag}(\lambda_1, ..., \lambda_d)$, one has that $\exp(-u/2) = \mathrm{diag}(e^{-\lambda_1/2}, ..., e^{-\lambda_d/2})$. Therefore, and using Equation (8), the coefficient $(k, l)$ of the matrix $\exp(-u/2)\Psi(E_{I_d,ij})\exp(-u/2)$ is:

$$
\begin{cases}
e^{-\lambda_i/2}e^{\lambda_i}e^{-\lambda_i/2} = 1 & \text{for } i = j = k = l, \\
\frac{1}{\sqrt{2}}e^{-\lambda_i/2}\frac{e^{\lambda_i}-e^{\lambda_j}}{\lambda_i-\lambda_j}e^{-\lambda_j/2} = \sqrt{2}\frac{\sinh\left(\frac{\lambda_i-\lambda_j}{2}\right)}{\lambda_i-\lambda_j} & \text{for } i \neq j \text{ and } (k,l) = (i,j) \\
0 & \text{if } (k,l) \neq (i,j).
\end{cases}
$$

Therefore, the matrix of $\Psi$ in the bases $(E_{I_d,ij})_{i\leq j}$ and $(E_{\exp(u),kl})_{k\leq l}$ is diagonal with diagonal coefficients 1 and $2\frac{\sinh\left(\frac{\lambda_i-\lambda_j}{2}\right)}{\lambda_i-\lambda_j}$. Thus, the determinant of $\Psi$ is the product of these coefficients, which gives the result for the diagonal case.

**Case 2:** $u \in T_{I_d}\mathcal{P}_d$ **general** Let us now consider the general case where $u \in T_{I_d}\mathcal{P}_d$ is not diagonal. One has that, as $u$ is symmetric, one can diagonalize it: $u = gdg^\top$ where $g$ is an orthogonal matrix and $d$ is diagonal. As $\exp(u) = g\exp(d)g^\top$, one has that $\mathrm{d}\exp(u) \cdot h = g\left(\mathrm{d}\exp(d) \cdot (g^\top hg)\right)g^\top$. Therefore, as we are only interested in the determinant of $\mathrm{d}\exp(u)$, and as $g$ are orthogonal, one has that $\det \mathrm{d}\exp(u) \cdot h = \det\left(\mathrm{d}\exp(d) \cdot (g^\top hg)\right)$. One thus need to compute $\mathrm{d}\exp(d) \cdot (g^\top hg)$, and using Dalechii-Krein formula, as in the diagonal case, one has:

$$\mathrm{d}\exp(d) \cdot (g^\top hg) = \left[\left[\exp^{[1]}(u)_{ij}\tilde{h}_{ij}\right]\right] \tag{9}$$

where $\exp^{[1]}(u)$ is defined as above and $g^\top hg = [[\tilde{h}_{ij}]]$. In order to do the same proof as for the diagonal case, one needs to modify the basis used in $T_{I_d}\mathcal{P}_d$ as in Equation (9), the coefficients of $g^\top hg$ appear (rather than directly the coefficients of $h$ as in the previous case). Therefore, we choose as basis for $T_{I_d}\mathcal{P}_d$ the basis $(E_{I_d,ij}^{(u)})_{ij}$ where $E_{I_d,ij}^{(u)} = gE_{I_d,ij}g^\top$. One can easily check that this basis is orthonormal and does not imply any volume changes. One can now use the same proof as for the diagonal case to compute the determinant of $\mathrm{d}\exp(u)$, which gives the result for the general case.

## D. The building block of the wrapped Gaussians

In this section, we give a proof of Proposition 4.5. For this, we will actually show a more precise proposition:

**Proposition D.1.** *Let $(p, \mu, \Sigma) \in \Theta$ and $X \sim \mathrm{WG}(p; \mu, \Sigma)$. Then,*

1. $p^{-1/2}Xp^{-1/2} \sim \mathrm{WG}(I_d; \mu, \Sigma)$,

2. $\mathrm{Exp}_p(\mathrm{Log}_p X - \mathrm{Vect}_p^{-1}(\mu)) \sim \mathrm{WG}(p; 0_{d(d+1)/2}, \Sigma)$,

3. $\mathrm{Exp}_p(\mathrm{Vect}_p^{-1}(\Sigma^{-1/2}\mathrm{Vect}_p(\mathrm{Log}_p X))) \sim \mathrm{WG}(p; \mu, I_{d(d+1)/2})$.

*Proof.* In this proof, we will only show the first two points of the above definition, the third one being similar to them.

1. Let $Y = p^{-1/2}Xp^{-1/2}$. We want to show that $Y \sim \mathrm{WG}(I_d; \mu, \Sigma)$. For this, let $\varphi\colon \mathcal{P}_d \to \mathbb{R}$ be a continuously bounded function. One has

$$\mathbb{E}[\varphi(Y)] = \int_{\mathcal{P}_d} \varphi(p^{-1/2}xp^{-1/2})f_{p;\mu,\Sigma}(x)\mathrm{dvol}(x)$$

$$= \int_{\mathcal{P}_d} \varphi(p^{-1/2}xp^{-1/2})\frac{1}{\sqrt{(2\pi)^d \det\Sigma}}\frac{\exp\left(-\frac{1}{2}(\mathrm{Vect}_p(\mathrm{Log}_p(x)) - \mu)^\top\Sigma^{-1}(\mathrm{Vect}_p(\mathrm{Log}_p(x)) - \mu)\right)}{|J_{I_d}(\log(p^{-1/2}xp^{-1/2}))|}\mathrm{dvol}(x).$$

Let us now define $\psi_p\colon x \mapsto p^{-1/2}xp^{-1/2}$. $\psi_p$ is a $\mathcal{C}^1$-diffeomorphism between $\mathcal{P}_d$ and $\mathcal{P}_d$. Moreover, as the volume element $\mathrm{dvol}$ is invariant by congruence of $\mathrm{GL}(d, \mathbb{R})$, the transformation $\psi_p$ does not imply any volume change. Therefore, by change of variables $y = \psi_p(x)$, one has:

$$\mathbb{E}[\varphi(Y)] = \int_{\mathcal{P}_d} \varphi(y)\frac{1}{\sqrt{(2\pi)^d \det\Sigma}}\frac{\exp\left(-\frac{1}{2}(\mathrm{Vect}_p(\mathrm{Log}_p(p^{1/2}yp^{1/2})) - \mu)^\top\Sigma^{-1}(\mathrm{Vect}_p(\mathrm{Log}_p(p^{1/2}yp^{1/2})) - \mu)\right)}{|J_{I_d}(\log(p^{-1/2}p^{1/2}yp^{1/2}p^{-1/2}))|}\mathrm{dvol}(y)$$

$$= \int_{\mathcal{P}_d} \varphi(y)\frac{1}{\sqrt{(2\pi)^d \det\Sigma}}\frac{\exp\left(-\frac{1}{2}(\mathrm{Vect}_p(p^{1/2}\log(y)p^{1/2}) - \mu)^\top\Sigma^{-1}(\mathrm{Vect}_p(p^{1/2}\log(y)p^{1/2}) - \mu)\right)}{|J_{I_d}(\log(y))|}\mathrm{dvol}(y).$$

Finally, using that $\mathrm{Vect}_p(p^{1/2}\log(y)p^{1/2}) = \mathrm{Vect}_{I_d}(p^{-1/2}p^{1/2}\log(y)p^{1/2}p^{-1/2}) = \mathrm{Vect}_{I_d}(\log(y))$ and $\mathrm{Log}_{I_d} = \log$, we have,

$$\mathbb{E}[\varphi(Y)] = \int_{\mathcal{P}_d} \varphi(y)\frac{1}{\sqrt{(2\pi)^d \det\Sigma}}\frac{\exp\left(-\frac{1}{2}(\mathrm{Vect}_{I_d}(\mathrm{Log}_{I_d}(y)) - \mu)^\top\Sigma^{-1}(\mathrm{Vect}_{I_d}(\mathrm{Log}_{I_d}(y)) - \mu)\right)}{|J_{I_d}(\log(y))|}\mathrm{dvol}(y).$$

This shows us that $Y \sim \mathrm{WG}(I_d; \mu, \Sigma)$.

2. Let now $Y = \mathrm{Exp}_p(\mathrm{Log}_p - \mathrm{Vect}_p^{-1}(\mu))$. We want to show that $Y \sim \mathrm{WG}(p; 0_{d(d+1)/2}, \Sigma)$. Let $\varphi \colon \mathcal{P}_d \to \mathbb{R}$ be a bounded continuous function. One has

$$\mathbb{E}[\varphi(Y)] = \int_{\mathcal{P}_d} \varphi(\mathrm{Exp}_p(\mathrm{Log}_p x - \mathrm{Vect}_p^{-1}(\mu))) f_{p;\mu,\Sigma}(x) \mathrm{dvol}(x)$$

$$= \int_{\mathcal{P}_d} \varphi(\mathrm{Exp}_p(\mathrm{Log}_p x - \mathrm{Vect}_p^{-1}(\mu)))$$

$$\times \frac{1}{\sqrt{(2\pi)^d \det \Sigma}} \frac{\exp\left(-\frac{1}{2}(\mathrm{Vect}_p(\mathrm{Log}_p(x)) - \mu)^\top \Sigma^{-1}(\mathrm{Vect}_p(\mathrm{Log}_p(x)) - \mu)\right)}{|J_p(\mathrm{Log}_p(x))|} \mathrm{dvol}(x).$$

Let us now define $\psi_p \colon x \mapsto \mathrm{Exp}_p(\mathrm{Log}_p x - \mathrm{Vect}_p^{-1}(\mu))$. $\psi_p$ is a $\mathcal{C}^1$-diffeomorphism between $\mathcal{P}_d$ and $\mathcal{P}_d$ and its inverse is $\psi_p^{-1} \colon y \mapsto \mathrm{Exp}_p(\mathrm{Log}_p x + \mathrm{Vect}_p^{-1}(\mu))$. By change of variables $y = \psi_p(x)$, one has:

$$\mathbb{E}[\varphi(Y)] = \int_{\mathcal{P}_d} \varphi(y) \frac{1}{\sqrt{(2\pi)^d \det \Sigma}} \frac{\exp\left(-\frac{1}{2} \mathrm{Vect}_p(\mathrm{Log}_p(y))^\top \Sigma^{-1} \mathrm{Vect}_p(\mathrm{Log}_p(y))\right)}{|J_p(\mathrm{Log}_p(y) + \mathrm{Vect}_p^{-1}(\mu))|} \frac{\mathrm{dvol}(y)}{|\det \mathrm{d}\psi_p(\psi^{-1}(y))|}.$$

We need to compute the change of volume term $\det \mathrm{d}\psi_p(\psi^{-1}(y))$. For this, let us start by saying that $\mathrm{d}\psi_p(x) = \mathrm{d}\mathrm{Exp}_p(\mathrm{Log}_p x - \mathrm{Vect}_p^{-1}(\mu)) \circ \mathrm{d}\mathrm{Log}_p x$ therefore, $\det \mathrm{d}\psi_p(x) = \det \mathrm{d}\mathrm{Exp}_p(\mathrm{Log}_p x - \mathrm{Vect}_p^{-1}(\mu)) \det \mathrm{d}\mathrm{Log}_p x$. Now using the fact that $\mathrm{d}\mathrm{Log}_p(y) = \left(\mathrm{d}\mathrm{Exp}_p(\mathrm{Log}_p(y))\right)^{-1}$ and the definition of $J_p(u) = \det \mathrm{d}\mathrm{Exp}_p(u)$ (see Theorem 4.2), we have

$$\det \mathrm{d}\psi_p(x) = J_p(\mathrm{Log}_p x - \mathrm{Vect}_p^{-1}(\mu)) \frac{1}{J_p(\mathrm{Log}_p x)}$$

and thus, plugging $\psi_p^{-1}(y)$ into the equation:

$$\det \mathrm{d}\psi_p(\psi_p^{-1}(y)) = J_p(\mathrm{Log}_p y) \frac{1}{J_p(\mathrm{Log}_p y + \mathrm{Vect}_p^{-1}(\mu))}.$$

Therefore,

$$\mathbb{E}[\varphi(Y)] = \int_{\mathcal{P}_d} \varphi(y) \frac{1}{\sqrt{(2\pi)^d \det \Sigma}} \frac{\exp\left(-\frac{1}{2} \mathrm{Vect}_p(\mathrm{Log}_p(y))^\top \Sigma^{-1} \mathrm{Vect}_p(\mathrm{Log}_p(y))\right)}{|J_p(\mathrm{Log}_p(y))|} \mathrm{dvol}(y).$$

This shows us that $Y \sim \mathrm{WG}(p; 0_{d(d+1)/2}, \Sigma)$.

3. For the third point, one can prove it similarly as the two previous one, having in mind that the vectorization $\mathrm{Vect}_p$ is an isometry (see Proposition A.4) therefore, neither $\mathrm{Vect}_p$ nor $\mathrm{Vect}_p^{-1}$ implies any volume changes.

$\square$

Therefore, Proposition 4.5 is a direct corollary of the previous result:

**Corollary D.2.** *Let* $X \sim \mathrm{WG}(I_d; 0_{d(d+1)/2}, I_{d(d+1)/2})$ *and let* $(p, \mu, \Sigma) \in \Theta$. *Let us define*

$$\Psi \colon x \in \mathcal{P}_d \mapsto p^{1/2} \mathrm{Exp}_p\left(\mathrm{Vect}_p^{-1}\left(\Sigma^{1/2}\left(\mathrm{Vect}_p \circ \mathrm{Log}_p x + \mu\right)\right)\right) p^{1/2}.$$

*Then,* $\Psi(X) \sim \mathrm{WG}(p; \mu, \Sigma)$.

## E. The wrapped Central Limit Theorem

In this section, we give a proof of the wrapped Central Limit Theorem stated at Theorem 4.7. For this, let $(X_i)_{i \in \mathbb{N}^*}$ be a sequence of independent and identically distributed random variables on the Riemannian manifold $\mathcal{P}_d$. We suppose that the sequence $(\mathrm{Vect}_{I_d}(\mathrm{Log}_{I_d}(X_i)))_{i \in \mathbb{N}^*}$ of random variables on $\mathbb{R}^{d(d+1)/2}$ satisfies the classical Central Limit Theorem. We want to show that the sequence $(X_i)_{i \in \mathbb{N}^*}$ satisfies the wrapped Central Limit Theorem.

*Remark* E.1. Let us start by saying that, as the map $x \mapsto \text{Vect}_{I_d} \circ \text{Log}_{I_d} x$ is a diffeomorphism between $\mathcal{P}_d$ and $\mathbb{R}^{d(d+1)/2}$, the sequence $(\text{Vect}_{I_d}(\text{Log}_{I_d}(X_i)))_{i \in \mathbb{N}^*}$ is also independent and identically distributed.

As the sequence $(\text{Vect}_{I_d}(\text{Log}_{I_d}(X_i)))_{i \in \mathbb{N}^*}$ satisfies the classical Central Limit Theorem in $\mathbb{R}^{d(d+1)/2}$, one has that,

$$\frac{1}{\sqrt{n}} \sum_{i=1}^{n} \left( \text{Vect}_{I_d}(\text{Log}_{I_d}(X_i)) - \mu \right) \xrightarrow[n\to\infty]{d} \mathcal{N}(0, \Sigma).$$

By defining $m = \text{Exp}_{I_d}(\text{Vect}_{I_d}^{-1}(\mu)) \in \mathcal{P}_d$, and using the linearity of $\text{Vect}_p$:

$$\text{Vect}_{I_d} \left( \frac{1}{\sqrt{n}} \sum_{i=1}^{n} (\text{Log}_{I_d}(X_i) - \text{Log}_{I_d}(m)) \right) \xrightarrow[n\to\infty]{d} \mathcal{N}(0, \Sigma).$$

Thus, by applying the continuous map $\text{Exp}_{I_d} \circ \text{Vect}_{I_d}^{-1}$ to the previous equation, by considering the fact that the convergence in distribution is stable by continuous maps (theorem 5.5 of Wasserman 2004) and the definition of the wrapped Gaussian (Definition 4.1), one has that

$$\text{Exp}_{I_n} \left( \frac{1}{\sqrt{n}} \sum_{i=1}^{n} (\text{Log}_{I_d}(X_i) - \text{Log}_{I_d}(m)) \right) \xrightarrow[n\to\infty]{d} \text{WG}(I_d; 0, \Sigma).$$

We can now simplify the left-hand side of the previous equation:

$$\text{Exp}_{I_n} \left( \frac{1}{\sqrt{n}} \sum_{i=1}^{n} (\text{Log}_{I_d}(X_i) - \text{Log}_{I_d}(m)) \right) = \exp\left( \frac{1}{\sqrt{n}} \sum_{i=1}^{n} (\log X_i - \log m) \right) \quad \text{using the expression of } \text{Exp}_{I_d} \text{ and } \text{Log}_{I_d} \text{ of Equation (3)}$$

$$= \exp\left( \sum_{i=1}^{n} (\log X_i + \log m^{-1}) \right)^{\frac{1}{\sqrt{n}}} \quad \begin{array}{l} \text{using } \log m = -\log m^{-1} \\ \text{and } \exp(\alpha x) = \exp(x)^\alpha \text{ for } \alpha \in \mathbb{R} \end{array}$$

$$= \exp\left( \sum_{i=1}^{n} \log(X_i \odot m^{-1}) \right)^{\frac{1}{\sqrt{n}}} \quad \text{using the definition of the logarithmic product}$$

$$= \left( \bigodot_{i=1}^{n} X_i \odot m^{-1} \right)^{\frac{1}{\sqrt{n}}}.$$

Therefore, one has the final result of the wrapped Central Limit Theorem:

$$\left( \bigodot_{i=1}^{n} X_i \odot m^{-1} \right)^{\frac{1}{\sqrt{n}}} \xrightarrow[n\to\infty]{d} \text{WG}(I_d; 0, \Sigma).$$

**The generalized wrapped CLT**  The previous version of the wrapped Central Limit Theorem is centered around the identity matrix $I_d$. However, one can generalize this theorem to any point $p \in \mathcal{P}_d$. For this, we need to introduce a generalized logarithmic product $\odot_p$ between two points $q_1, q_2 \in \mathcal{P}_d$:

$$q_1 \odot_p q_2 = \text{Exp}_p(\text{Log}_p q_1 + \text{Log}_p q_2).$$

In the same way as before, one can show the following generalized wrapped CLT

$$\left( \bigodot_{i=1}^{n} {}_p X_i \odot_p M^{-1} \right)^{\frac{1}{\sqrt{n}}} \xrightarrow[n\to\infty]{d} \text{WG}(p; 0, \Sigma).$$

# F. An extension to wrapped Elliptically Contoured Distributions

As one can see from the expression of the density of a wrapped Gaussian $\mathrm{WG}(p; \mu, \Sigma)$ given at Theorem 4.2, it is intrinsically linked to the density of the multivariate Gaussian $\mathcal{N}(\mu, \Sigma)$. This suggests a possible extension to *Elliptically Contoured Distributions* (chapter 6 of Johnson (1987) or Delmas et al. 2024). We recall the definition of an Elliptically Contoured distribution:

**Definition F.1** (Elliptically Contoured Distribution). A random vector $X \in \mathbb{R}^d$ follows an Elliptically Contoured distribution if there exists $\mu \in \mathbb{R}^d$, $\Sigma \in \mathcal{P}_d$ and a function $g$ such that $X$ has density

$$f_X(x) = k \det(\Sigma)^{-1/2} g\left((x - \mu)^\top \Sigma^{-1}(x - \mu)\right)$$

where $k$ is a normalizing factor. We denote $X \sim \mathrm{EC}(\mu, \Sigma, g)$.

For example, the multivariate Gaussian $\mathcal{N}(\mu, \Sigma)$ is an Elliptically Contoured distribution with $g \colon t \mapsto \exp(-t/2)$. Another example is the multivariate t-distributions for which $g \colon t \mapsto (1 + t/\nu)^{-d+\nu/2}$ (see V.B of chapter 1 of Delmas et al. 2024). One can then extend what has been done previously on the wrapped Gaussian to define *Wrapped Elliptically Contoured Distributions* just like above:

**Definition F.2** (Wrapped Elliptically Contoured). Let $p \in \mathcal{P}_d$, $\mu \in \mathbb{R}^{d(d+1)/2}, \Sigma \in \mathcal{P}_{d(d+1)/2}$ and $g$ be a function. Then, a random vector $X$ on $\mathcal{P}_d$ follows a Wrapped Elliptically Contoured denoted $\mathrm{WEC}(p; \mu, \Sigma, g)$ if

$$X = \mathrm{Exp}_p(\mathrm{Vect}_p^{-1}(\mathbf{t})), \ \mathbf{t} \sim \mathrm{EC}(\mu, \Sigma, g).$$

One can then compute the density of $\mathrm{WEC}(p; \mu, \Sigma, g)$ similarly as in Theorem 4.2. Moreover, all the work done on the equivalence relation for wrapped Gaussians stays valid for wrapped elliptically contoured distributions.

# G. The proofs on the equivalence relation

In this section, we want to give proofs of the different results of Section 4.4. We recall the propositions and give their proofs. Let us start by Proposition 4.9.

**Proposition G.1.** *Let* $(p, \mu, \Sigma) \in \Theta$ *and* $t \in \mathbb{R}$. *One has that* $\mathrm{WG}(p; \mu, \Sigma)$ *and* $\mathrm{WG}(e^t p; \mu - t\nu, \Sigma)$ *are equal where* $\nu = \mathrm{Vect}_p(p) = (1, \cdots, 1, 0, \cdots, 0) \in \mathbb{R}^{d(d+1)/2}$.

*Proof.* In this following, we denote by $\gamma$ the function $\gamma \colon t \mapsto e^t p$. Let us denote by $\tilde{f}$ the density of $\mathrm{WG}(\gamma(t); \mu - t\nu, \Sigma)$ and by $f$ the density of $\mathrm{WG}(p; \mu, \Sigma)$. We want to show that $\tilde{f} = f$. Let $x \in \mathcal{P}_d$, by Theorem 4.2, one has:

$$\tilde{f}(x) = \frac{1}{\sqrt{(2\pi)^d \det \Sigma}} \frac{\exp\left(-\frac{1}{2}\left(\mathrm{Vect}_{\gamma(t)}(\mathrm{Log}_{\gamma(t)}(x)) - \mu + t\nu\right)^\top \Sigma^{-1}\left(\mathrm{Vect}_{\gamma(t)}(\mathrm{Log}_{\gamma(t)}(x)) - \mu + t\nu\right)\right)}{|J_{\gamma(t)}(\mathrm{Log}_{\gamma(t)}(x))|}.$$

One has, that

$$\begin{aligned}
\mathrm{Log}_{\gamma(t)}(x) &= \gamma(t)^{1/2} \log(\gamma(t)^{-1/2} x \gamma(t)^{-1/2}) \gamma(t)^{1/2} \\
&= e^t p^{1/2} \log(e^{-t} p^{-1/2} x p^{-1/2}) p^{1/2} \quad \text{using that } \gamma(t) = e^t \\
&= e^t p^{1/2} \log(e^{-t} I_d) p^{1/2} + e^t \mathrm{Log}_p(x) \quad \text{using that } e^{-t} I_d \text{ and } p^{-1/2} x p^{-1/2} \text{commute} \\
&= -t e^t p + e^t \mathrm{Log}_p(x).
\end{aligned}$$

Furthermore, one has:

$$\begin{aligned}
\mathrm{Vect}_{\gamma(t)}(\mathrm{Log}_{\gamma(t)}(x)) &= -t e^t \mathrm{Vect}_{\gamma(t)}(p) + e^t \mathrm{Vect}_{\gamma(t)}(\mathrm{Log}_p(x)) \quad \text{using the linearity of } \mathrm{Vect}_{\gamma(t)} \\
&= -t e^t \mathrm{Vect}_{I_d}(\gamma(t)^{-1/2} p \gamma(t)^{-1/2}) + e^t \mathrm{Vect}_{I_d}(\gamma(t)^{-1/2} \mathrm{Log}_p(x) \gamma(t)^{-1/2}) \\
&= -t e^t e^{-t} \mathrm{Vect}_{I_d}(p^{-1/2} p p^{-1/2}) + e^t e^{-t} \mathrm{Vect}_{I_d}(p^{-1/2} \mathrm{Log}_p(x) p^{-1/2}) \\
&= -t \mathrm{Vect}_p(p) + \mathrm{Vect}_p(\mathrm{Log}_p(x)).
\end{aligned}$$

Therefore, the numerator of the density $\tilde{f}$ can be rewritten as:

$$\exp\left(-\frac{1}{2}(\text{Vect}_p(\text{Log}_p(x)) - \mu)^\top \Sigma^{-1}(\text{Vect}_p(\text{Log}_p(x)) - \mu)\right).$$

which is the same numerator as $f$.

Let us now focus on the denominator. One has:

$$
\begin{aligned}
J_{\gamma(t)}(\text{Log}_{\gamma(t)}(x)) &= J_{I_d}(\gamma(t)^{-1/2}\,\text{Log}_{\gamma(t)}(x)\gamma(t)^{-1/2}) \\
&= J_{I_d}(e^{-t}p^{-1/2}\,\text{Log}_{\gamma(t)}(x)p^{-1/2}) \\
&= J_{I_d}(-tI_d + p^{-1/2}\,\text{Log}_p(x)p^{-1/2}) \quad \text{using the computation of } \text{Log}_{\gamma(t)}(x).
\end{aligned}
$$

We recall that the Jacobian determinant of the exponential map at the identity is:

$$J_{I_d}(u) = 2^d \prod_{i<j} \frac{\sinh\left(\frac{\lambda_i(u) - \lambda_j(u)}{2}\right)}{\lambda_i(u) - \lambda_j(u)}$$

where $(\lambda_i(u))_i$ are the eigenvalues of $u$. Moreover, the eigenvalues of $u := -\alpha t I_d + p^{-1/2}\,\text{Log}_p(x)p^{-1/2}$ are

$$\lambda_i(u) = -\alpha t + \lambda_i\left(p^{-1/2}\,\text{Log}_p(x)p^{-1/2}\right).$$

Thus, for all $i < j$, one has:

$$\lambda_i(u) - \lambda_j(u) = \lambda_i\left(p^{-1/2}\,\text{Log}_p(x)p^{-1/2}\right) - \lambda_j\left(p^{-1/2}\,\text{Log}_p(x)p^{-1/2}\right)$$

and therefore, this leads to:

$$J_{I_d}(u) = J_{I_d}\left(p^{-1/2}\,\text{Log}_p(x)p^{-1/2}\right) = J_p\left(\text{Log}_p(x)\right).$$

So the denominator of the density $\tilde{f}$ is the same as the denominator of the density $f$ and therefore, the two densities are equal. □

*Remark* G.2. The function $\gamma \colon t \mapsto e^t p$ is actually the geodesic with initial point $p$ and initial velocity $p$. Indeed, the expression of the geodesic $\Gamma_{q,V}(t)$ with initial point $q$ and initial velocity $V \in T_q\mathcal{P}_d$ is (see Pennec 2020):

$$\forall t \in \mathbb{R}, \ \Gamma_{q,V}(t) = q^{1/2}\exp(tq^{-1/2}Vq^{-1/2})q^{1/2}.$$

Therefore, the geodesic $\gamma$ with initial point $p$ and initial velocity $p$ (which is a symmetric matrix, therefore an element of $T_p\mathcal{P}_d \simeq \mathcal{S}_d$) is:

$$\forall t \in \mathbb{R}, \ \Gamma_{p,p}(t) = e^t p = \gamma(t).$$

We now want to show Proposition 4.14 that we recall underneath:

**Proposition G.3.** *Let* $\theta = (p; \mu, \Sigma) \in \Theta$ *be a tuple of parameters. Then, the minimal representative of the class* $[\theta]$ *as defined at Definition 4.13 is* $\theta^{min} = (p^{min}; \mu^{min}, \Sigma^{min})$ *where*

$$
\begin{cases}
p^{min} = e^{\frac{1}{d}\sum_{i=1}^d \mu_i}p, \\
\mu^{min} = \mu - \dfrac{1}{d}\sum_{i=1}^d \mu_i \nu, \\
\Sigma^{min} = \Sigma.
\end{cases}
$$

*Proof.* We want to find the smallest $\mu^{\min}$ in the sens of $\|\cdot\|_2$ and the corresponding $p^{\min}$ such that $(p; \mu, \Sigma) \cong (p^{\min}; \mu^{\min}, \Sigma^{\min})$. As all the $\mu$ in the equivalence class of $[\theta]$ are of the form $\mu - t\nu$ for $t \in \mathbb{R}$, to find the smallest $\mu^{\min}$, one needs to minimize the following function:

$$\varphi \colon t \mapsto \|\mu - t\nu\|_2^2 = \|\mu\|_2^2 - 2t\langle \mu, \nu \rangle + t^2\|\nu\|_2^2.$$

One thus has:
$$\varphi'(t) = -2\langle \mu, \nu \rangle + 2t\|\nu\|_2^2.$$

The minimum is reached at $t^{\min} = \frac{\langle \mu, \nu \rangle}{\|\nu\|_2^2}$ with $\|\nu\|_2^2 = d$ and $\langle \mu, \nu \rangle = \sum_{i=1}^d \mu_i$. Therefore, one has:

$$\begin{cases} p^{\min} = e^{\frac{1}{d} \sum_{i=1}^d \mu_i} p \\ \mu^{\min} = \mu - \frac{1}{d} \sum_{i=1}^d \mu_i \nu = \left( \mu_1 - \frac{1}{d} \sum_{i=1}^d \mu_i, \cdots, \mu_d - \frac{1}{d} \sum_{i=1}^d \mu_i, \mu_{d+1}, \cdots, \mu_{d(d+1)/2} \right). \end{cases}$$

$\square$

## H. Why does estimating $p$ using the Riemannian mean fails in the general case?

We said in Section 5 that when $\mu^\star \neq 0$, using the Riemannian mean $\mathfrak{G}(x_1, ..., x_N)$ does not work to estimate the parameters $(p^\star, \mu^\star, \Sigma^\star)$. Let us explain why. For this, we suppose in the following that $\mu^\star \neq 0$. Let $\hat{p}_N$ be the Riemannian mean: $\hat{p}_N = \mathfrak{G}(x_1, ..., x_N)$. Then, we can use Proposition 5.1 to compute the MLE of $\mu$ and $\Sigma$:

$$\hat{\mu}_N = \frac{1}{N} \sum_{i=1}^N \text{VLog}_{\hat{p}_N}(x_i),$$

$$\hat{\Sigma}_N = \frac{1}{N} \sum_{i=1}^N \left( \text{VLog}_{\hat{p}_N}(x_i) - \hat{\mu}_N \right) \left( \text{VLog}_{\hat{p}_N}(x_i) - \hat{\mu}_N \right)^\top.$$

where we recall that $\text{VLog}_{\hat{p}_N}$ is the vectorization at $\hat{p}_N$ of $\text{Log}_{\hat{p}_N}$ i.e. $\text{VLog}_{\hat{p}_N} = \text{Vect}_{\hat{p}_N} \circ \text{Log}_{\hat{p}_N}$. Let us focus on $\hat{\mu}_N$. Using the linearity of $\text{Vect}_{\hat{p}_N}$, we can write that

$$\hat{\mu}_N = \text{Vect}_{\hat{p}_N} \left( \frac{1}{N} \sum_{i=1}^N \text{Log}_{\hat{p}_N}(x_i) \right).$$

According to proposition 3.4 of Moakher (2005), as $\hat{p}_N$ is the Riemannian mean of the points $(x_1, ..., x_N)$, we have the following:

$$\sum_{i=1}^N \text{Log}_{\hat{p}_N}(x_i) = 0.$$

Therefore, $\hat{\mu}_N = 0$. It is therefore not a good estimator of $\mu^\star \neq 0$. That is why we do not use the Riemannian mean as an estimator of $p$ in a general setting when we do not know *a priori* that $\mu^\star = 0$.

## I. More details on the MLE experiments

In this section, we give more details on the setup of the synthetic experiments lead in Section 5 to assess the quality of the estimation of parameters of a wrapped Gaussian using an MLE. Upon acceptation, we will release the code used to perform these experiments. To obtain the results plotted at Figure 3, we repeated 5 times the experiment with different true parameters $\theta^\star = (p^\star, \mu^\star, \Sigma^\star)$ randomly generated. Here are details on how we generated the true parameters:

- For $p^\star$, we use the function `generate_random_spd_matrix` from the library PyRiemann (Barachant et al., 2024). This function generates a random SPD matrix by generating a random matrix $A$ and then computing $\exp((\bar{X} + s * A)^\top (\bar{X} + s * A))$ where $\bar{X}$ and $s$ are parameters chosen by the user. We set $\bar{X} = 0.1I_d$ and $s = 1$.

- For $\mu^\star$, we generate a random vector of size $d(d+1)/2$ with values in $[0, 0.1]$.

- For $\Sigma^\star$, we generate a random SPD matrix using the same function as for $p^\star$ with $\bar{X} = 0.01I_{d(d+1)/2}$ and $s = 0.02$.

We chose relatively small values for $\bar{X}$ and $s$ because otherwise, when the dimension $d$ is large, the generated parameters are very far from identity leading to numerical instability.

*Figure 4.* Comparison of the estimation of the parameters of a wrapped Gaussian when the covariance matrix $\Sigma$ is diagonal or full. The dimension of the SPD matrices in this experiment is $d = 10$.

## J. Estimating the parameters of a wrapped Gaussian when the covariance matrix $\Sigma$ is diagonal

In this section, we detail the experiments on the estimation of the parameters of a wrapped Gaussian when the covariance matrix $\Sigma$ is diagonal. We used the same setup as in the previous experiments detailed in Appendix I except that we generated $\Sigma^\star$ as a diagonal matrix. The diagonal was uniformly sampled in $[0, 1]$. We repeated the experiment 5 times with different true parameters. The goal for this experiment was to assess the impact of the structure of the covariance matrix on the estimation of the parameters. We wanted to show that when the covariance matrix is diagonal, the estimation of the parameters requires fewer samples. We plotted the results in the case of dimension $d = 10$ at Figure 4. One can see that, while the estimation of $p$ and $\mu$ is not affected by the structure of $\Sigma$, the estimation of $\Sigma$ is better when $\Sigma$ is diagonal. With a lot less samples, one can achieve a significantly better estimation of $\Sigma$ when it is diagonal. This is coherent as the number of coefficients to estimate is reduced when $\Sigma$ is diagonal. Therefore, this setting of $\Sigma$ diagonal can be a good choice when the number of samples is limited.

## K. More details on the real data experiments

Let us start by giving more details on the different datasets used in the experiments and the preprocessing we used.

- **BCI Datasets**: We considered 2 datasets from Brain Computer Interfaces (BCI) for our experiments: *BNCI2014001* (Leeb et al., 2007) and *Zhou2016* (Zhou et al., 2016). They consist of several subjects and several sessions per subjects doing a Motor Imagery task (Pfurtscheller & Neuper, 2001). We used the library MOABB (Aristimunha et al., 2023) to load and preprocess the data. For each EEG, We start by applying a standard band-pass filter with range $[7; 35]$ Hz. Then, we used the Ledoit-Wolf shrunk covariance matrix (Ledoit & Wolf, 2004) to in order to compute the covariance matrices and to avoid ill-conditioned matrices. The experiment we lead was cross-subject: each classifier was trained on all subject except one and tested on this last subject.

- **AirQuality**: This dataset is from the Beijing Municipal Monitoring Center. It is a dataset of air quality monitored from 34 different sites in Beijing, China (Hua et al., 2021). For each site, six atmospheric pollutants where recorded every hour: $CO$, $NO_2$, $O_3$, $PM_{10}$, $PM_{2.5}$ and $SO_2$. We used the same preprocessing as in Smith et al. (2022) to get a point cloud of 102 covariance matrices of size $6 \times 6$. Each covariance matrix has a label depending on which period it represents: weekdays, weekends or holidays.

- **Indiana, Pavia Uni, Salinas**: These three dataset of hyperspectral remote sensing datasets are all publicly available at https://www.ehu.eus/ccwintco/index.php/Hyperspectral_Remote_Sensing_Scenes. Each dataset contains one hyperspectral image of a certain region containing a unique number of reflectance bands. We applied the same preprocessing that in Collas et al. (2021) or (Bouchard et al., 2024) that consists in four main steps. First, we normalize the data by subtracting the image global mean. Then, we apply a PCA to reduce the dimension of the data to 5. A sliding window with no overlap is then used around each pixel for data sampling and then vectorized. In our experiments, we used a window of size $25 \times 25$. Finally, we compute the covariance matrix of each vectorized window using the Sample Covariance Matrix to get a point cloud of covariance matrices. For each covariance matrix, its class was computed by taking the majority class of the pixels in the window.

- **Textile**: This dataset is made of a set of real images from textile manufacturing that contain non-defective and defective woven textiles. These images come from the public MVTec Anomaly Detection dataset (Bergmann et al., 2021). The same preprocessing as in Smith et al. (2023) was applied to get the covariance matrices. The two classes correspond to defective and non-defective textiles.

- **BreizhCrops**: This dataset is intended for supervised classification of field crops from satellite time series. We used the dataset `FRH01` that is composed of satellite time series from the French region *Finistère*. As they are multivariate time-series, we simply converted them to covariance matrices using the Oracle Approximating Shrinkage estimator (Chen et al., 2010). The classes correspond to different types of crops. More details on this dataset can be found in the original paper (Rußwurm et al., 2020).

For the non-BCI datasets (AirQuality, Indiana, Pavia Uni, Salinas, Textile and BreizhCrops), we used a 5-fold cross-validation to evaluate the performance of the classifiers.

Let us also give some details on the implementation of the different classifiers used in the experiments.

- The MDM is implemented using the library PyRiemann (Barachant et al., 2024).

- The TS-LDA uses the `TangentSpace` class from PyRiemann (Barachant et al., 2024) and the LDA from Scikit-learn (Pedregosa et al., 2011).

- The TS-QDA uses the `TangentSpace` class from PyRiemann (Barachant et al., 2024) and the Naive Bayes classifier from Scikit-learn (Pedregosa et al., 2011).

- For the Ho-WDA and He-WDA, we implemented them using our MLE to estimate the parameters of the wrapped Gaussians. To optimize the MLE, we used in practice the Riemannian Conjugate Gradient method (Boumal, 2023) with a maximum of $1,000,000$ iterations and a max time set to 2 hours.

