# OpenReview forum: "Wrapped Gaussian on the manifold of Symmetric Positive Definite Matrices"
_ICML.cc/2025/Conference — ICML 2025 poster_

### Official Review · Reviewer_PgG4 · 2025-02-26

**Overall Recommendation:** 3

**Summary:**

This paper studies the non-isotropic wrapped Gaussian distribution on the manifold of positive definite (PD) matrices. Specifically, the authors derive theoretical properties of the non-isotropic wrapped Gaussian distribution and propose maximum likelihood estimators for its parameters. They also define an equivalence relation between the set of parameters of two wrapped Gaussian and resolve the non-identifiability issue of the wrapped Gaussian for PD matrices. Finally, the authors provide new interpretations to several known classifiers on PD matrices through the lens of wrapped Gaussian distributions.

**Claims And Evidence:**

Overall, all the claims and results in this paper are supported by rigorous proofs and/or simulation results. However, I am not quite convinced by the claim on the second column of Page 8 (Line 403) that "we do not observe a clear dominance of the He-WDA over the Ho-WDA". The major issue of this claim is that the Monte Carlo experiments are only repeated for 5 times, which is clearly not enough. It should be at least 100 times. Moreover, it is not intuitive about why the He-WDA behaves worse than the Ho-WDA on many of data examples. Shouldn't Ho-WDA be a special case of He-WDA when all the covariance matrices for different classes are the same.

There are also some minor issues in the paper that I pointed out below.

**Essential References Not Discussed:**

To the best of my knowledge, the paper did a good job in discussing the related works. The only main concern is the novelty of this paper when compared with the prior works that consider the centered distribution on the tangent space. Intuitively, if we know the tangent space of a manifold, it does not seem to be very difficult to center the data and/or distribution with respect to the origin of the tangent space. I encourage the authors to address this concern in more detail. Appendix H could be one example, but still not convincing enough.

**Experimental Designs Or Analyses:**

Yes, I have checked the validity of experimental analyses. The only concern is the Monte Carlo repetition times that I mentioned above.

**Methods And Evaluation Criteria:**

The proposed methods and evaluation criteria basically makes sense for the wrapped Gaussian distribution problem and its related classification problems. As a side note, since the authors mentioned that they can sample from the wrapped Gaussian distribution, it would be better to outline the sampling procedure in the main paper or Appendix. Specifically, a procedure without rejection sampling is expected.

**Other Comments Or Suggestions:**

1. In the Abstract (Line 21-25), the sentence "We introduced a non-isotropic wrapped Gaussian by leveraging the exponential map, we derive theoretical properties of this distribution and propose a maximum likelihood framework for parameter estimation." is not correct in grammar.

2. Second line in Section 4.2: a typo "pus-forward".

3. Proposition 4.9: it should be clearly stated how many ones are there in $\nu=(1,...,1,0,...,0)$. The same applies to Proposition 4.14.

4. I am confused about Remark 4.12: can we just set $\mu_{\alpha}= -t  \mathrm{Vect}_{p{\alpha}}(p{\alpha})$ so that the equivalence class can contain $\mu=0$?

5. Figure 3: For the case $d=10$, why is it almost consistently better than the case when $d=5$ for MLE to estimate $p^*$? It seems that $d=10$ is a more challenging problem. Does it mean that the Riemannian conjugate gradient algorithm doesn't fit to this problem?

6. The first column of Page 6 (line 308-310): can we center the data and then apply the method of moment? what difficulties are there for preventing this straightforward adaptation?

7. The first column of Page 7 (Line 366): typo "arex pulled back.."

8. In the conclusion part, it seems to me that it may not be feasible to extend all the classical machine learning models that rely on Gaussian distributions to the manifold of SPD matrices. The computational issues incurred by calculating the exponential and logarithmic maps are huge barriers.

9. Why did the authors present all the proofs in tiny font? It is hard for readers to read them.

10. Line 793 on Page 15: typo: "does dot imply...".

11. Line 812: "$Id$" at the end of the equation should be $I_d$.

12. Second point in Proposition D.1: the random variable $X$ is missing. It should be $\mathrm{Log}_p X$.

13. Line 860 on Page 16: typo "diffeomorphisme".

14. Line 1061 on Page 20: If $||\nu||_2^2 = n d$,

then where does the "n" go in the expression of $p_{\min}$?

**Other Strengths And Weaknesses:**

The writing and proofs of this paper are of good standard.

The only main concern is the novelty of this paper when compared with the prior works that consider the centered distribution on the tangent space. Intuitively, if we know the tangent space of a manifold, it does not seem to be very difficult to center the data and/or distribution with respect to the origin of the tangent space. I encourage the authors to address this concern in more detail. Appendix H could be one example, but still not convincing enough.

**Questions For Authors:**

Please see my concerns and comments above. I am happy to increase my scores if the authors can carefully address my comments and concerns.

**Relation To Broader Scientific Literature:**

To my knowledge, the wrapped distributions have been studied in directional statistics dating back to at least 1970s. This paper extends this wrapped technique for the Gaussian distribution, or more generally, elliptically contoured distributions, to the manifold of positive definite matrices. As the authors pointed out, wrapped distributions have been studied on homogeneous Riemannian manifolds, but using a different techniques as the one proposed in this paper. More relatedly, some exponential-wrapped distributions on symmetric spaces were studied in the literature as well, but as the authors in this paper pointed out, these related works consider the distribution on the tangent space to always be centered. In this paper, the authors consider a slightly more general setting with the wrapped Gaussian distribution on the manifold of positive definite matrices not necessarily being centered.

**Theoretical Claims:**

I have checked all the proofs and results in both the main paper and supplementary materials.

---

> ### Author Rebuttal · Authors · 2025-03-31
>
> First, we would like to thank the reviewer for their work, their valuable comments and interesting questions.
>
> ## Regarding the "Claims And Evidence":
>
> We agree that the He-WDA should be a special case of the Ho-WDA when the covariance matrices for each classe are the same. In order to evaluate the performance of the He-WDA and Ho-WDA, we used a 5-fold cross-validation on each dataset. This is classicly done in ML to evaluate pipelines. We were limited by the amount of data in some datasets. Moreover, as the He-WDA estimates one covariance matrix per class, it has more parameters to estimate and then requires a lot of data per class. If the number of samples per class is low, the estimation of the covariance matrix can be very noisy and the performance of the He-WDA can be worse than that of the Ho-WDA. For example, for the dataset Salinas, some classes have only a few hundred samples, which is not enough to estimate the covariance matrix accurately. The Ho-WDA, on the other hand, estimates a single covariance matrix for all the classes and is less sensitive to the number of samples per class. We will add this explanation in the final version of the paper.
> ## Regarding the "Methods And Evaluation Criteria":
>
> The sampling procedure of a wrapped Gaussian is very simple and does not rely on any sophisticated rejection sampling algorithm. Indeed, to sample from $WG(p;\mu,\Sigma)$, one can simply sample a point $x$ from the Euclidean Gaussian $\mathcal{N}(\mu,\Sigma)$ and then project the samples on the manifold $P_d$ by $Exp_p(Vect^{-1}_p(x))$. The exponential map $Exp_p$ as well as the vectorization $Vect_p$ are both simple to compute. We will add this algorithm in the final version of the paper.
>
> ## Regarding the "Other Strengths And Weaknesses":
>
> We would like to emphasize the fact that allowing the distribution to be non-centered in the tangent space is not just a “slightly more general setting''. For more details on this point, please refer to our answer to point 3 of reviewer 9rYP.
>
> ## Regarding the "Other comments or suggestions":
> We will correct the different typos highlighted in this review.
>
> 3. If the SPD matrices are of size $d \times d$ (like we consider in the paper), then $\nu$ is the concatenation of $d$ ones and $d(d+1)/2-d = d(d-1)/2$ zeros. We will clarify this point in the final version.
> 4. This is an important point. Let us reformulate remark 4.12: Let us consider a wrapped Gaussian $WG(p;\mu, \Sigma)$. Then, the equivalent wrapped Gaussians are of the form $WG(e^t p; \mu + tVect_p(p), \Sigma)$ for $t \in R$. If $\mu$ and $Vect_p(p)$ are aligned i.e., there exists $\tilde{t}$ such that $\mu = -\tilde{t}Vect_p(p)$, then the equivalence class contains a wrapped Gaussian with $\mu = 0$. However, as $\mu$ and $Vect_p(p) = (1,\cdots,1,0,\cdots,0)$ (the concatenation of $d$ ones and $d(d-1)/2$ zeros, see previous point) are not aligned (for example, take $\mu = \nu + (1,\cdots,0) = (2,1,\cdots,1,0,\cdots,0)$), then there exists no $t$ such that $\mu = -tVect_p(p)$ and the equivalence class does not contain a wrapped Gaussian with $\mu = 0$. We will clarify this point in the final version.
> 5. We agree with you that we could expect that the problem with $d=10$ is more challenging than the problem with $d=5$. According to this, we would expect the error on the estimation of $p^\star$ to be higher for $d=10$ than for $d=5$. However, the results of Figure 3 show the contrary. Maybe we need to repeat more than 5 times the experiment to have a more accurate estimation of the error. We will try to increase the number of repetitions in the final version of the paper.
> 6. In order to center the data on the tangent space (to have $\mu^\star = 0$), one needs to know $p^\star$ and $\mu^\star$ (more details on how to center the data are given in appendix D).  Indeed, we need to know the tangent plan in which we want to center the data. As here our goal is to estimate the parameters, we do not know $p^\star$ and $\mu^\star$ and thus cannot center the data. Moreover, if one simply “centers'' the data using the Riemannian mean $\mathfrak{G}(x_1,...,x_N)$ , one does not have any guarantees that the “centered” data will have $\mu = 0$.
> 8. Computing exponential and logarithmic maps remains a bottleneck in SPD matrix geometry. However, a trade-off may exist between computational cost and performance gains. Theoretically, Euclidean Gaussian-based methods extend to $P_d$ via our wrapped Gaussian, though practical challenges will arise in applications, requiring careful choices.
> 9. We will modify the font size of the proofs to make them more readable.
> 12. Indeed, the random variable $X$ is missing in the second point of Proposition D.1. We will correct this error.
> 14. The $n$ in $||\nu||_2^2 = nd$ is a typo. Since $\nu$ consists of $d$ ones and $d(d-1)/2$ zeros, its squared norm is simply $d$. Therefore, the expression for $p_min$ is correct. This error will be fixed.

---

> > ### Comment · Reviewer_PgG4 · 2025-04-08
> >
> > The paper has its merit, but the contributions are not extremely groundbreaking. I will keep my score at somewhere between 3-4.

---

### Official Review · Reviewer_nsaj · 2025-03-04

**Overall Recommendation:** 4

**Summary:**

The authors propose a new version of Gaussian on the SPD manifold (with the affine metric) by wrapping a Gaussian from a tangent space onto the manifold. Their method has two main differences than previously proposed methods 1. the distribution need not be isotropic and 2. the footpoint of the distribution on the manifold need not be the mean of the distribution. The authors further extend the ideas of statistical classification using their wrapped Gaussian distribution.

This review has been updated after the rebuttal period.

**Claims And Evidence:**

Almost all their claims are supported.

The only claims which I am not entirely sure are on the section 6.1 claims of MLE. These claims do seem intuitive, but I think a more thorough justification is necessary.

**Essential References Not Discussed:**

NA

**Experimental Designs Or Analyses:**

Yes, all were checked.

**Methods And Evaluation Criteria:**

Yes, the authors test their methods on both simulated data and real data. The simulated data is somewhat low dimensional with a max d of 10 however the real data is on the same order so it seems appropriate.

**Other Comments Or Suggestions:**

In the introduction, I think the authors should mention they are considering the affine metric earlier as there are two natural options for a metric on this space.

Second page I think there is a hanging sentence "the authors work on symmetric spaces." the sentence seems out of place or incomplete.

"In the sequel" seems odd choice of wording.



Typos and errors:
4.1 "tangent plan" -> "tangent plane"
4.2 starts with "pus-forward"
5.1 the theta star parameters missing comma
6.1 "They rely on metric that arex pulled" -> "They rely on metrics that are pulled"
All paper: Inconsistent notation of $WG(p,\mu,\Sigma$ v $WG(p;\mu,\Sigma)$
Figure 3, the third panel has the star in the wrong location

**Other Strengths And Weaknesses:**

See comments below.

**Questions For Authors:**

In section 5 it's unclear when it is assumed $\Sigma$ is diagonal. It seem in 5.1 that it will be always but then in 5.2 it seems as if only sometimes.

Is the normalizing constant in MDM dependent on the mean on the distribution?

When describing LDA and QDA the authors mention that "all training points are sent to the tangent space via the exponential map" should this not be the log map?


In section 5.1 is the $p^2$ a typo?

**Relation To Broader Scientific Literature:**

The SPD manifold is often used but the structure is often ignored as it can be difficult to handle, the authors propose a method on the manifold leveraging the structure. Many authors have considered other distributions on SPD such as Hajri et al. who proposed the Laplace and this paper seems like a direct comparison in terms of impact.

**Theoretical Claims:**

Only glanced at most proofs but took a longer look at the proof in section H of the appendix.

---

> ### Author Rebuttal · Authors · 2025-03-31
>
> First, we would like to thank the reviewer for their work, their valuable comments and interesting questions.
>
> ## Regarding the Claims and Evidence:
> You say that you have doubts about the claims on the MLE made in section 6.1. The only claim on the MLE made in this section is that the LDA uses an MLE on the training data to learn the parameters of each class. Are you referring to this claim? If it is the case, we refer to section 4.3 of [1] (page 109) where the formulas for the estimated parameters are given for the Euclidean LDA. On can check that they coincide with the formula of the MLE in the Euclidean Gaussian case.
> ## Regarding the Comments Or Suggestions:
> 1. We will mention earlier in the introduction that we focus on the affine-invariant metric on the manifold of SPD matrices and mention that other choices could be made (for example Log-Euclidean metric).
> 2. We will check and correct the different typos and errors throughout the paper.
> ## Regarding the Questions For Authors:
> 1. In section 5, $\Sigma$ is never assumed to be diagonal. The experiments lead in the section were always led with a full $\Sigma$. However, we mention that one can assume that $\Sigma$ is diagonal to simply the estimation problem (reducing the number of parameters to estimate from $O(d^4)$ to $O(d^2)$ where $d$ is size of the SPD matrices). This assumption of a diagonal $\Sigma$ is done in the experiments led in Section 6.1 and in Appendix J. We will clarify this point in the final version.
> 2. The normalizing constant in the MDM does not depend on the mean of the distributions. Indeed, as shown in Proposition 1 of [2], the normalizing constant of the isotropic Gaussian distribution depends only on $\sigma$, not on the mean.
> 3. Absolutely, in the description of the Tangent Space LDA and QDA, one should read “logarithm map” instead of “exponential map”. We will correct this error.
> 4. In section 5.1, the $p^2$ is not meant to be $p$ squared but the $^2$ refers to a footnote. This can indeed be confusing and will be corrected.
>
> Finally, you mention paper [3] that completely fits in the scope of our paper, and we will make sure to add this reference to our section 2 on related works.
>
> [1] Hastie, T., Tibshirani, R., and Friedman, J. The Elements of Statistical Learning. Springer Series in Statistics. Springer, New York, NY, 2009.
>
> [2] Said, S., Hajri, H., Bombrun, L., and Vemuri, B. C. Gaussian Distributions on Riemannian Symmetric Spaces: Statistical Learning With Structured Covariance Matrices. IEEE Transactions on Information Theory, 64 (2):752–772, February 2018.
>
> [3]  Hatem Hajri, Ioana Ilea, Salem Said, Lionel Bombrun, Yannick Berthoumieu. Riemannian Laplace distribution on the space of symmetric positive definite matrices. 2015

---

### Official Review · Reviewer_9rYP · 2025-03-16

**Overall Recommendation:** 1

**Summary:**

The authors proposed a wrapped Gaussian formalism and give ML estimator for mean and covariance. The authors showed usefulness of proposed formulation in LDA and QDA on several datasets.

**Claims And Evidence:**

1. My biggest concern regarding the intrinsic nature of the formalism as claimed, the formalism is essentially a extrinsic formulation on tangent space, define Gaussian on the tangent space and push-forward on the manifold

**Essential References Not Discussed:**

1. Gaussian distributions on Riemannian symmetric spaces

**Experimental Designs Or Analyses:**

The experiments are sound, although not convincing how much benefit we get using wrapped Gaussian

**Methods And Evaluation Criteria:**

Yes

**Other Comments Or Suggestions:**

Please see weakness section

**Other Strengths And Weaknesses:**

Minor comments:

The nations are uncommon, for example why use both vector and matrix using Lower case?!
In all metrics on SPD, it has non-positive curvature, not just affine-invariant metric: “Once
endowed with this metric, Pd is a complete connected Riemannian manifold of non-positive curvature” — this statement can be clarified.Also all Riemannian metrics give same sign of curvature, as the metrics are equivalent in that sense.
3. Section 3.1 reads dense and a laundry list of things, better to clarify saying why (if possible) (and where) you need this to use this. All of these are well-known and well-studied across geometers so the non-familiar reader should know in which section you will use these tools.
4. In section 4.2, “pus-forward” should be “push-forward”.
5. In section 4.2, “The Jacobian determinant” better call it determinant of the Jacobian.

Major Comments:

1. Whenever one defines something on Eucliden space and pull over manifold the intrinsic nature is gone. The authors argued in Introduction how classical tools are non-intrinsic: “However, classical Eu-
clidean probability distributions fail to capture the intrinsic geometry of the underlying manifold.” So I am confused why authors definition of wrapped Gaussian is non-intrinsic as in section 4.1.

2. The entire argument of being intrinsic doesn’t hold as everything defined in relate of wrapped Gaussian is extrinsic including Therorem 4.2, proposition 4.3

3. In section 4.2., the authors said they allow usage of \mu, which is very trivial extension of centered Gaussian.

4. Why in section 4.3, the CLT will be something novel to prove? As it depends on CLT on the tangent space. Am I missing something here?

5. The estimation of ML estimators of mean and std follows from Euclidean Gaussians, so again section 5 follows naturally from Euclidean MLEs.

6. The performance of LDA and QDA using wrapped Gaussian is not very superior over the competition as in Table 2. So not sure why we will use such a formalism even if we ignore the extrinsic nature of it.

**Questions For Authors:**

Please see weakness.

**Relation To Broader Scientific Literature:**

The ideas are relevant for the community.

**Theoretical Claims:**

Yes, the claims hold.

---

> ### Author Rebuttal · Authors · 2025-03-31
>
> ## Regarding the “Essential References Not Discussed”
> The reference [1] you mention is actually the first reference given in Section 2 devoted to related works, on the first page of our paper. It was a key reference for the development of our theory as the authors proposed an __isotropic__ Gaussian distribution Riemannian Symmetric Spaces (e.g. SPD). Our goal was to study a __non-isotropic__ Gaussian distribution on SPD matrices. This is clearly stated in the first 6 lines of Section 2.
>
> ## Regarding the "Other Strengths and weaknesses":
> ### Minor comments:
> Regarding the choice of geometry, not all metrics behave the same on the manifold of SPD matrices. We could have chosen a flat metric leading to a Euclidean space.
> ### Major comments:
> 1. Although we only used the word "intrinsic" twice in our paper, it may have confused our reader. There is a trade off between working with an anisotropic distribution and having a fully intrinsic distribution. Indeed, as we detail in Section 2, Pennec has defined in [2] a purely anisotropic and intrinsic Gaussian distribution on a Riemannian manifold. However, as detailed in section 2:
> 	- The normalizing constant is expressed by an integral over the whole manifold $P_d$ (intractable in practice).
> 	- It uses a concatenation matrix rather than a covariance matrix, and the relation between those two matrices requires the computation of an integral over $P_d$.
> 	- No sampling method has been studied for it.
> 	Hence, we tried to take the best of both world in our work by tackling the anisotropy and to provide an easy-to-sample distribution, and we completed these objectives.
> 2. The Gaussian distribution defined in [1] also relys (indirectly) on the choice of a particular tangent space. Indeed, in the p.d.f. of the Gaussian, the difference between a given point $y$ and the mean $\bar{x}$ is computed using $Log_{\bar{x}}y$ (differenc in the tangent space). However, this distribution is called “intrinsic”.
> 3. Allowing the distribution to be non-centered in the tangent space raises several theoretical and practical issues that are not present when $\mu = 0$:
> 	- The non-identifiability issue raise by the choice of $\mu \neq 0$ is solved in section 4.4.
> 	- As stated in remark 4.12, the equivalence classes of a given wrapped Gaussian (WG) does not necessarily contain a centered WG. Therefore, the choice of $\mu$ has a real impact on the expressivity of our distribution.
> 	- The estimation of the parameters is another issue: if one assumes $\mu = 0$, then the estimation of the parameters is straightforward (by the methods of moments as it has been done by Chevallier et al. (2022)). However, if $\mu \neq 0$, then the estimation of the parameters is more complex and requires the use of a full MLE.
> 	- We would like to recall that, unlike the Euclidean case, $\mu$ does not model the mean of the distribution, as we introduce a new parameter $p$. Maybe you got confused between those two parameters.
>
> 	Given these precisions, could you argue what is specifically trivial and why ?
> 4. The CLT on the tangent space is not novel to prove. However, the CLT on the manifold was (to the best of our knowledge) never stated in the literature. We believe that this result is interesting as it shows the interest of considering WG on SPD matrices as a WG appears as the asymptotic distribution of a logarithmic product of i.i.d. random variables on the manifold. If this result is known, could you give a reference ?
> 5. The estimation of the parameters of a WG is not a trivial extension of the Euclidean Gaussian case. As we explain in proposition 5.1, __in the special case where $p^\star$ is known a priori__, the MLE of $\mu$ and $\Sigma$ are the same as the Euclidean case. However, in the general case, the MLE of $p$ does not have a closed-form solution and is dependent on the MLE of ($\mu$, $\Sigma$). Thus, the estimation of the three parameters $p$, $\mu$ and $\Sigma$ must rely on the full maximum likelihood optimization problem stated in section 5.1. Once again, could you specify what is trivial here ?
> 6. Using WG for classification with LDA and QDA primarily served to illustrate its applications. Our goal was to demonstrate how the theory developed earlier enables the construction of machine learning tools directly on the manifold of SPD matrices. The main value of WG lies in offering a novel way to model data on SPD matrices. Additionally, in Section 6.2, we showed how classical SPD classifiers can be reinterpreted through the WG framework, providing a unifying probabilistic perspective. We believe that introducing such a perspective is valuable, and this paper represents a first step in that direction.
>
> [1] Said, S. et al, Gaussian Distributions on Riemannian Symmetric Spaces: Statistical Learning With Structured Covariance Matrices. IEEE Trans. Inf. Theory, 2018.
>
> [2] Pennec, X. Intrinsic Statistics on Riemannian Manifolds: Basic Tools for Geometric Measurements. J. Math. Imaging Vis, 2006.

---

### Decision · Program_Chairs · 2025-05-01

**Decision:**

Accept (poster)

**Comment:**

This paper is borderline, with a split between reviewers, all of whom are seasoned experts on this topic.

Those that liked the paper praise its results, particularly that the SPD manifold occurs often and is difficult to handle, indicating a need to develop better theoretical tools for working with it. Reviewers also felt the paper covered the literature well, and there were no complaints about presentational issues in spite of the fact that such complaints are extremely common.

The reviewer who was more critical, to my understanding, was primarily concerned with whether it is valid or not to describe the proposed method as intrinsic. I am not convinced this criticism matters to a decisive degree: whether a method is formulated in an extrinsic or intrinsic manner is secondary to whether or not it is a useful tool for practitioners to use on their problems. This reviewer also did not respond to a request for comment in the discussion phase.

On basis of this comparison, I believe this work's merits outweigh its downsides, and recommend a weak accept.